# Mode of intracontinental mountain building controlled by lower crustal composition and mantle lithosphere depletion

Xi Xu [1,2,3,4,5] ✉, Andrew V. Zuza [6], Taras Gerya [4], Lin Chen [7], Xingtao Kuang[2], Hanlin Chen [5] ✉, Baodi Wang[2], Jingao Liu [8], Xuhua Shi [2], Yanyun Sun[2], Lei Wu [5], Song Han[2], Xiubin Lin [5], Shufeng Yang[5] & An Yin[3]

Tectonic plate convergence is accommodated across the continental lithosphere via discrete lithospheric subduction or distributed shortening and thickening. These end-member deformation modes control intra-plate mountain building, but their selection mechanism remains unclear. The variable composition of the continental crust and lithospheric mantle, which impacts its density and rheology, can be inferred by the distribution of magnetic-indicated crustal iron. Here we demonstrate that vertically coherent pure-shear shortening dominated the active Tian Shan orogen, central Asia, based on high-resolution aeromagnetic imaging and geophysical-geodetic observations. Integrating these findings with thermomechanical collisional models reveals that the mode of intracontinental deformation depends on contrasts in lower crust composition and mantle lithosphere depletion between the converging continents and central orogenic region. Distributed shortening prevails when the converging continents have a more iron-enriched mafic crust and iron-depleted mantle lithosphere when compared to the intervening orogenic region. Conversely, continental subduction occurs without such lithospheric contrasts. This result explains how the Tian Shan orogen formed via distributed lithospheric thickening without continental subduction or underthrusting. Our interpretations imply that iron distribution in the crust correlates with lithospheric compositional, density, and rheological structure, which impacts the preservation and destruction of Earth's continents, including long-lived cratons, during intracontinental orogeny.

Despite its success in quantifying the kinematics of rigid oceanic plates, classical plate tectonic theory fails to account for the widespread occurrence of the distributed continental deformation controlled by the non-rigid behavior of the continents[1]. Continental deformation, best expressed by intracontinental mountain building and exemplified by the development of the Tibetan plateau and Tian Shan in the Cenozoic Indo-Asia collision zone[1,2], may be accommodated by the following three end-member modes: (1) discrete simple-shear orogeny with focused deformation at the continental subduction or underthrusting interface[3–8], (2) distributed pure-shear orogeny with shortening strain distributed across the entire orogen[6,9–11], or (3) ductile flow in the middle and/or lower crust causing crustal thickening or thinning[12,13]. Geodynamic modeling in the past decades shows that the mode of continental deformation is dictated by the thermal state and mechanical strength of the continental lithosphere[2,10,14–16]. Although the iron content in the continental crust

and the mantle lithosphere impacts its density distribution[17,18] and viscous rheology[19], the role of iron enrichment and depletion in controlling continental deformation has not been systematically assessed.

The iron concentration within the continental lithosphere depends on the degree of mafic-ultramafic melt extraction during partial melting of the mantle lithosphere protolith; more melt depletion of the protolith may result in a less dense and thicker mantle lithosphere[17,20,21]. Thereby, the partitioning of lithospheric iron predominantly occurs during the formation of oceanic and/or continental crust, which in time may become spatially disconnected with the respective melt-depleted continental lithosphere[20,21]. As the result of multi-stage formation of the continental lithosphere[20,21], the composition of its crustal and mantle layers can be broadly variable, which affects its rheological and density structure and thus its tectonic response to plate convergence processes. The distribution of iron in the continental lithosphere can therefore be used as one of the key indicators of regional variations in lithospheric composition and rheological structure. However, quantifying the iron distribution in a continental lithosphere is challenging because the occasionally available xenoliths sample only a small fraction of the continental lithosphere below[18] with limited spatial resolution. Fortunately, iron enrichment and its distinct magnetic signature in the crust can be detected by aeromagnetic anomalies, and the associated iron distribution in the continental lithosphere can be quantified using inverse methods[22].

To leverage this knowledge and examine the role of iron redistribution in controlling continental deformation, here we investigated the lithospheric structure of the Tian Shan mountain belt, central Asia, which lies ~1500 km to the north of the Cenozoic India-Asia plate boundary (Fig. 1a). The Tian Shan is a well-known example of intracontinental orogeny that is actively deforming due to the far-field impacts of India-Asia convergence[1,5]. The mountain range is bounded by the Tarim and Kazakhstan blocks[5,6], to the south and north,

respectively. The Tian Shan orogen initiated in the Early Paleozoic, following the long-lived accretion of a series of microcontinents and island arcs and the subduction of the Paleo-Asian Ocean[23,24]. After Triassic-Jurassic tectonic quiescence[25,26], the Tian Shan experienced localized uplift and erosion during the Cretaceous-Paleocene[27], accelerated deformation in the Miocene[28], and is still being actively shortened today[29]. Crustal shortening is evenly distributed across the orogen as revealed by geodesy[30,31], Quaternary deformation studies[29], and crustal seismicity[24]. Shear-wave splitting analysis of core-refracted XKS waves shows the fast orientations of lithospheric anisotropic fabrics parallel the trend of the Tian Shan orogen[6]. Receiver function studies show a relatively shallow Moho beneath the Kazakhstan and Tarim blocks at ~40–50 km depth that becomes deeper underneath the Tian Shan at ~60–70 km depth[4,5]. The Moho is imbricated by a series of thrusts beneath the Tian Shan[5]. Therefore, Cenozoic crustal deformation has been distributed across the Tian Shan during progressive India-Asia convergence[32].

Here, we show how iron enrichment in the lower crust of two orogen-bounding continents is correlated with the pure-shear mode of deformation across the 1500 km-long and 200–300 km-wide active Tian Shan belt in central Asia (Fig. 1). Our numerical modeling demonstrates that such correlation can be explained by the presence of a depleted iron-poor mantle lithosphere that is rheologically coupled to the iron-rich mafic lower crust that bounds the two sides of the Tian Shan belt, the latter of which is characterized by more felsic iron-poor lower crust and less depleted, thinner mantle lithosphere.

## Results

Figure 2a shows a high-resolution total-field aeromagnetic anomaly map of the Tian Shan and its adjacent regions (Supplementary Figs. S1–S3; "Methods"). The aeromagnetic imaging reveals a series of long-wavelength, high-amplitude positive anomalies distributed

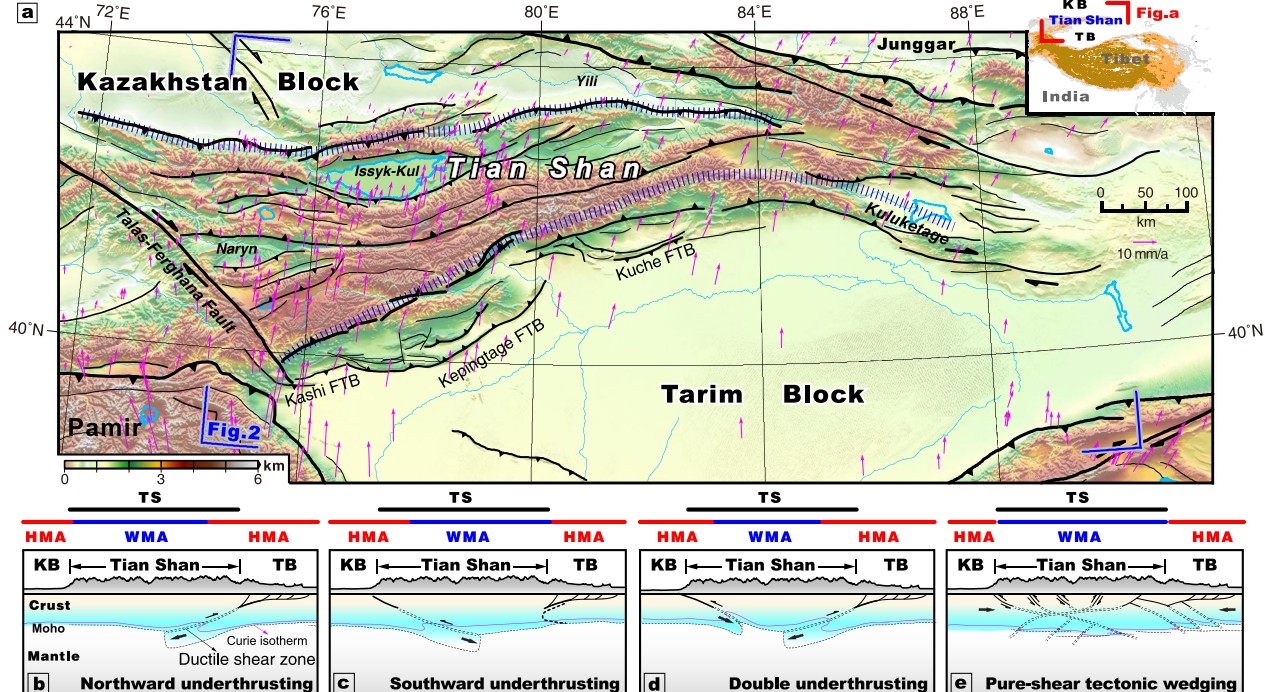

**Fig. 1 | Tectonic map of the Tian Shan and proposed tectonic models for its development.** (**a**) Map of the Tian Shan and its bounding Tarim and Kazakhstan blocks. Pink arrows show global positioning system (GPS) velocities in and around the Tian Shan with respect to stable Eurasisa[30]. Dotted wide-blue lines indicate the locations of the southern and northern margins of Tian Shan; FTB, fold-thrust belt. (**b–e**) Proposed models for the development of the Tian Shan, including (**b**) Tarim block underthrusting northward beneath the Tian Shan[7,33]. (**c**) Kazakhstan block underthrusting southward beneath the Tian Shan[4,6], (**d**) double-sided block unthrusting beneath the Tian Shan[5,8,16,34,35], and (**e**) pure-shear tectonic wedging with no appreciable underthrusting (this study). HMA highly positive magnetic anomaly. WMA weakly negative magnetic anomaly. TB Tarim block. KC Kazakhstan block. TS Tian Shan topographic range.

across the Tarim and Kazakhstan blocks, and more diffuse short-wavelength signals are generally observed within the Tian Shan interior (Fig. 2a and Supplementary Fig. S3). The northern Tarim block is characterized by long-wavelength, high-amplitude, positive magnetic highs (such as labeled A–C and circled-H in Fig. 2a) and neutral magnetic (or reversely magnetized) lows, such as along the northern margin of the Tarim craton (N in Fig. 2a). This magnetic character changes abruptly to the north at the Southern Tian Shan thrust belt (Fig. 2a), where the interior of the Tian Shan orogen is characterized by negative magnetic anomalies except for a few local short-wavelength magnetic highs. Farther north across the Northern Tian Shan fault, the Kazakhstan block is also dominated by long-wavelength, high-amplitude, positive magnetic highs (labeled D and E in Fig. 2a). The longer wavelength signals observed in the craton-like blocks are lower crustal-scale features, whereas the shorter wavelength anomalies in the Tian Shan are shallower (upper crust) (Supplementary Figs. S4 and S5; "Methods").

## Discussion

### Pure-shear orogeny across the Tian Shan

The actively deforming Tian Shan, bounded by the Kazakhstan and Tarim blocks (Fig. 1a) and formed by reactivating a Paleozoic orogen[5,23] during the Cenozoic India-Asia collision, may have developed via (1) northward[7,33] (Fig. 1b), southward[4,6] (Fig. 1c) or double-sided[5,8,16,34,35] (Fig. 1d) continental subduction/underthrusting, or (2) thick-skinned distributed pure-shear shortening[11,29,31] (Fig. 1e). Pure-shear orogeny predicts strain distributed across the entire orogen, whereas simple-shear predicts focused deformation at the continental subduction/underthrusting interface. Aeromagnetic imaging can test intracontinental simple-versus pure-shear orogeny because magnetic signals and anomaly patterns of the converging continents should project into the deformed orogens if simple-shear continental subduction is occurring (Fig. 1b–d). For example, this test was originally performed in the Canadian Rocky Mountains by Price[36], who showed that the North American craton was underthrust below the orogen and the coeval Cordilleran magmatic arc, as the unique cratonic continental magnetic anomalies can be projected beneath both (Supplementary Fig. S8a).

The high-amplitude, long-wavelength, positive magnetic anomalies (labeled A-E in Fig. 2a, b) characteristic of the Tian Shan-bounding cratonic blocks terminate along the northern and southern edges of the Tian Shan (Fig. 2c, d). This magnetic pattern refutes models of underthrusting or subduction of the bounding blocks beneath the Tian Shan, which predicts that the characteristic magnetic signals (i.e., high-amplitude, long-wavelength, positive anomalies) of the bounding blocks are observed within the Tian Shan orogen (Fig. 1b–d). Moreover, the thickened Tian Shan crust is not driving channel flow outward into the Tarim or Kazakhstan continents, which would result in magnetic signals from the Tian Shan projecting into the bounding continents. Therefore, the magnetic pattern revealed in this study and comprehensive geophysical-geodetic observations suggests that vertically coherent pure-shear thickening and shortening dominated the entire Tian Shan orogen (Fig. 1e). This pure-shear mode of lithospheric deformation is supported by the present knowledge of distributed Quaternary shortening across the range[29], diffuse Mesozoic-Cenozoic thermochronology ages across the Tian Shan implying distributed slow exhumation despite fast strain rates[37], widespread shallow crustal seismicity[24] (Supplementary Fig. S9a), and a linear deceleration of the velocity field across the entire orogen[30,31] (Supplementary Fig. S9b, S9c). Moreover, structural reinterpretation of recent receiver function profiles[4,5] are consistent with lithospheric tectonic wedging structures bounding the Tian Shan, characterized by multiple offsets of Moho (Supplementary Fig. S10 and Fig. 3). Relocated seismicity

below the north-dipping, south-directed thin-skinned Kepingtage thrust belt[33] shows a prominent south-dipping seismic zone at ~20–40 km depth (Supplementary Fig. S11) that we interpret as a tectonic wedge (Fig. 3), approximately consistent with the presence of a sharp Moho offset at ~40–50 km depth revealed by a receiver function transect across the westernmost segment of the Tian Shan[5].

### Mafic lower crust and depleted mantle lithosphere drives pure-shear orogeny

A compilation of global lower crust samples shows an inverse correlation between iron and silica contents regardless of age (Fig. 4). Therefore, the high-amplitude, long-wavelength, positive magnetic anomalies of the Tarim and Kazakhstan blocks reveal that their lower crust must be more mafic than the Tian Shan orogen. Crustal scale inversions of the aeromagnetic anomaly reveal that the interpreted mafic bodies are voluminous and thick (>25 km) in blocks' lower crust (Supplementary Figs. S4 and S5; "Methods"), indicative of a high-volume percentage of magnetite ($Fe_3O_4$) (Fig. 2b and Supplementary Figs. S6 and S7; "Methods"). In contrast, the upper crust of the Tian Shan orogen and the bounding blocks is characterized by extensive high-frequency magnetic patches (Supplementary Figs. S6 and S7) that correspond to isolated mafic magmatic bodies. Seismic and tomographic studies have imaged high-velocity crustal layers across the Tarim block, interpreted as mafic zones embedded in the middle-lower crust[38]. Conversely, the Tian Shan orogen, which involved Paleozoic accretion of island arcs[23], should have a more felsic iron-poor lower crust possibly formed by mafic-crust delamination and felsic lower crust relamination during the arc-related crust-mantle differentiation process[39]. Globally, the presence of a high-Vp and highly magnetic lower crust within rigid continental blocks and the occurrence of a low-Vp and weakly magnetic lower crust in Paleozoic-Mesozoic orogens suggests that underplating of mantle-generated mafic melts may have predominantly occurred in cratonic continental areas and not in recent orogens[40,41].

Voluminous mafic rocks in the lower crust contribute to higher mechanical strength for the orogen-bounding cratonic blocks compared with the Tian Shan orogen[42]. Furthermore, the cratonic continents should have a relatively old, thick and depleted mantle lithosphere[17,20,21], which will likely contrast with the younger, thinner and more fertile lithospheric mantle of the Tian Shan orogen. Therefore, we argue that high-amplitude, long-wavelength magnetic anomalies (Figs. 2a, 3 and 4) reveal a lithosphere with a predominantly mafic lower crust underlain by a more depleted (i.e., Mg-rich and Fe-poor) buoyant, rheologically strong, and thick cratonic lithospheric mantle[17,18] (Fig. 3b, c). The observed global correlation of the rheological strength and thickness of the continental lithosphere with its age[2,43,44] is consistent with ancient cratons having a more mafic lower crust and a coupled refractory mantle keel.

Accordingly, we propose that such positively buoyant, iron-depleted cratonic mantle supports the overlying rigid iron-rich mafic crust to act as a stiff indenter to partition strain localization in the Tian Shan lithosphere with a relatively weak, felsic crust and coupled iron-enriched (i.e., less depleted and rheologically weaker) lithospheric mantle (Fig. 3). Iron depletion in the Tarim and Kazakhstan lithospheric mantle inhibits their subduction, which in turn favors pure-shear construction of the Tian Shan orogen (Fig. 3).

Similar to the Tarim and Kazakhstan cratonic continents, other low-relief cratonic blocks located within and around the Tibetan plateau also exhibit long wavelength and high amplitude magnetic patterns, such as the Qaidam, Sichuan, Ordos, and Junggar basins (Supplementary Fig. S12). The spatial correspondence between high-amplitude, long-wavelength, positive magnetic signals and the lack of intracontinental deformation supports our interpretation that the aeromagnetic signals

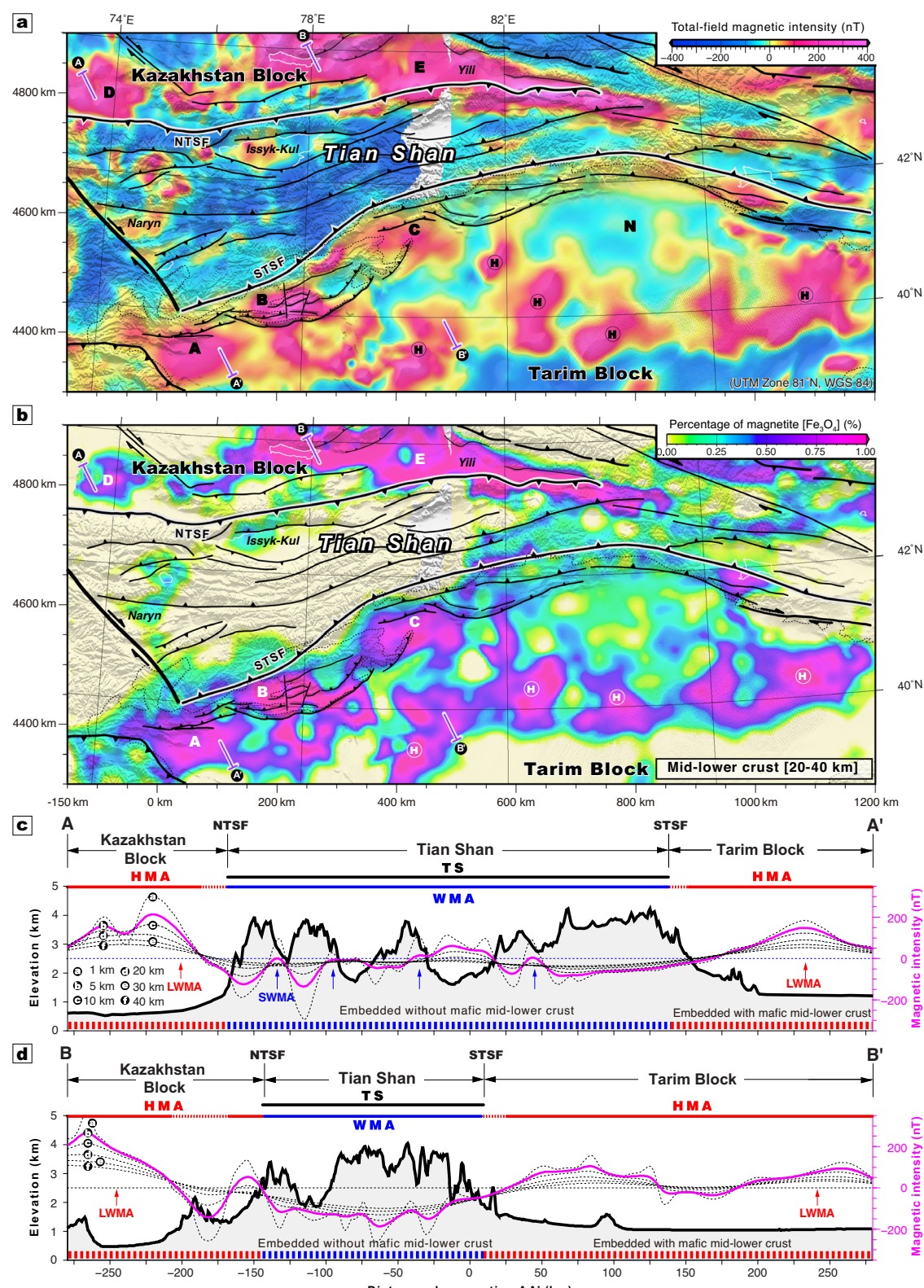

**Fig. 2 | Magnetic anomalies across the Tian Shan and its surrounding regions.**
(**a**) 5 km upward-continuation magnetic anomaly map across the Tian Shan and its bounding blocks. (**b**) Average volume percentage of magnetite (Fe$_3$O$_4$) in the lower crustal rock (at depths from 20 to 40 km) (Supplementary Fig. S6). (**c**) Magnetic anomaly and (**d**) topography across the analyzed profiles AA' and BB', respectively. Red and blue arrows represent the locations of long-wavelength and short-wavelength magnetic anomalies, LWMA and SWMA, respectively. The SWMA indicates upper-crustal, magnetite-bearing igneous intrusions. Circled a to f represent the altitude of upward continuation for the aeromagnetic anomalies (Supplementary Fig. S4). The lower crust of Tian Shan and its bounding cratonic continents are indicated by weakly-negative and highly-positive magnetic anomalies, WMA and HMA, respectively. Circled-H indicates the long-wavelength, high-amplitude, positive magnetic high. TS Tian Shan topographic range. NTSF Northern Tian Shan fault. STSF Southern Tian Shan fault.

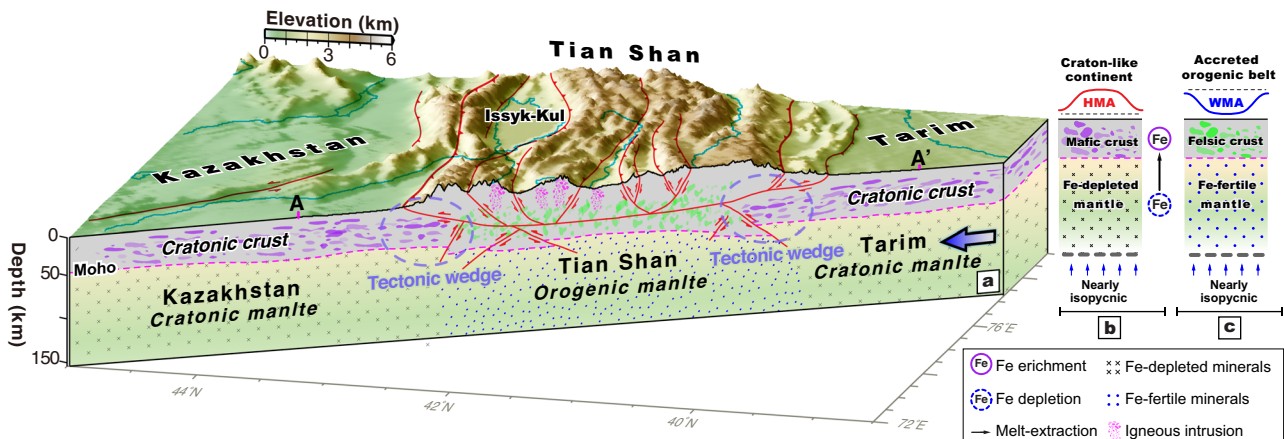

**Fig. 3 | Tectonic model for Cenozoic Tian Shan orogeny.** (**a**) lithospheric-scale tectonic model for the formation of the Tian Shan and a conceptual lithospheric iron-partitioning model comparing (**b**) craton-like versus (**c**) orogenic lithosphere and corresponding magnetic pattern predictions. Fault (red lines) and Moho discontinuities (dashed purple lines) are reinterpreted for the joint inversion receiver function profiles[4,5] (AA', Supplementary Fig. S10). Aeromagnetic imaging shows lower crustal iron-enrichment of the cratonic lithosphere as long-wavelength positive highly-magnetic anomalies (HMA), which we interpret to be compositionally coupled with melt-driven iron-depletion from the mantle lithosphere, and lower crustal iron-depletion of accreted orogenic lithosphere as negative weakly-magnetic anomalies (WMA), corresponding to underlying iron-fertile mantle lithosphere. The large arrow indicates the tectonic stress derived from the ongoing Indo-Asia convergence.

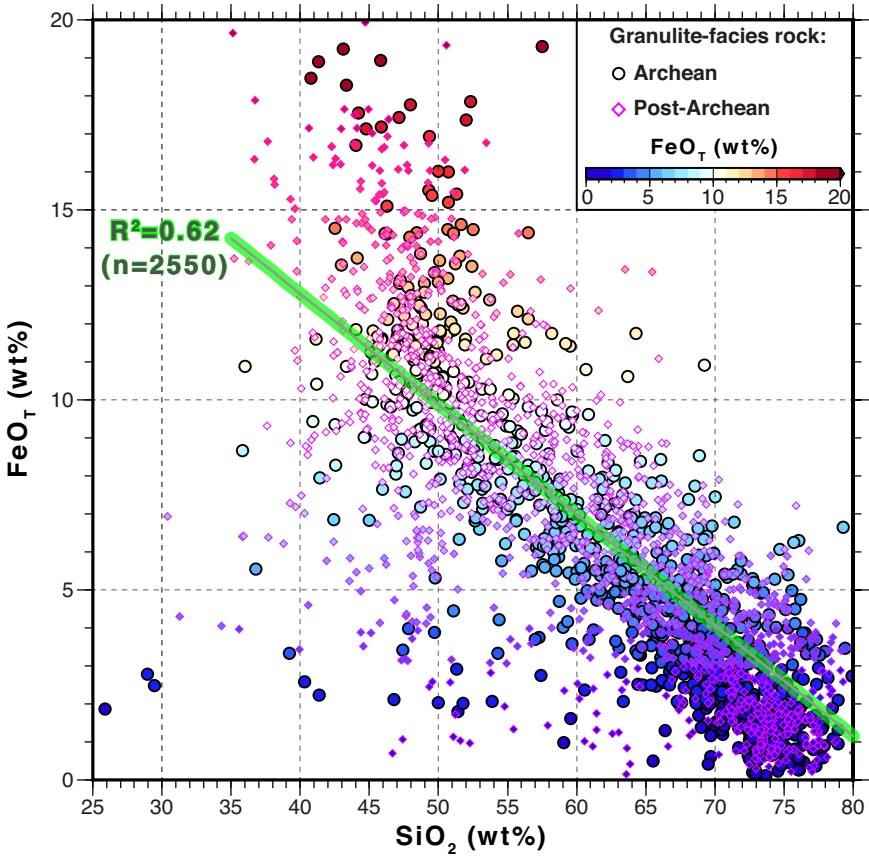

**Fig. 4 | Continental lower crustal FeO$_T$ and SiO$_2$ correlation.** FeO$_T$ versus SiO$_2$ in the continental granulite-facies (lower-crustal) rocks, where FeO$_T$ represents all Fe taken as FeO, derived from the database of rock compositions from Archean and post-Archean terrains[39]. The green line shows the linear best fit of the inverse correlation between FeO$_T$ and SiO$_2$.

may be diagnostic for identifying iron distribution in the lithosphere, with an iron-enriched lower mafic crust compositionally coupled with a depleted mantle. This configuration would result in a strong, rigid crust and buoyant, thick mantle lithosphere that would shield the crust from substantial deformation. We hypothesize that lower crustal composition and mantle lithosphere depletion revealed by iron distribution may control the mode of intracontinental deformation between pure- and simple-shear styles of orogeny.

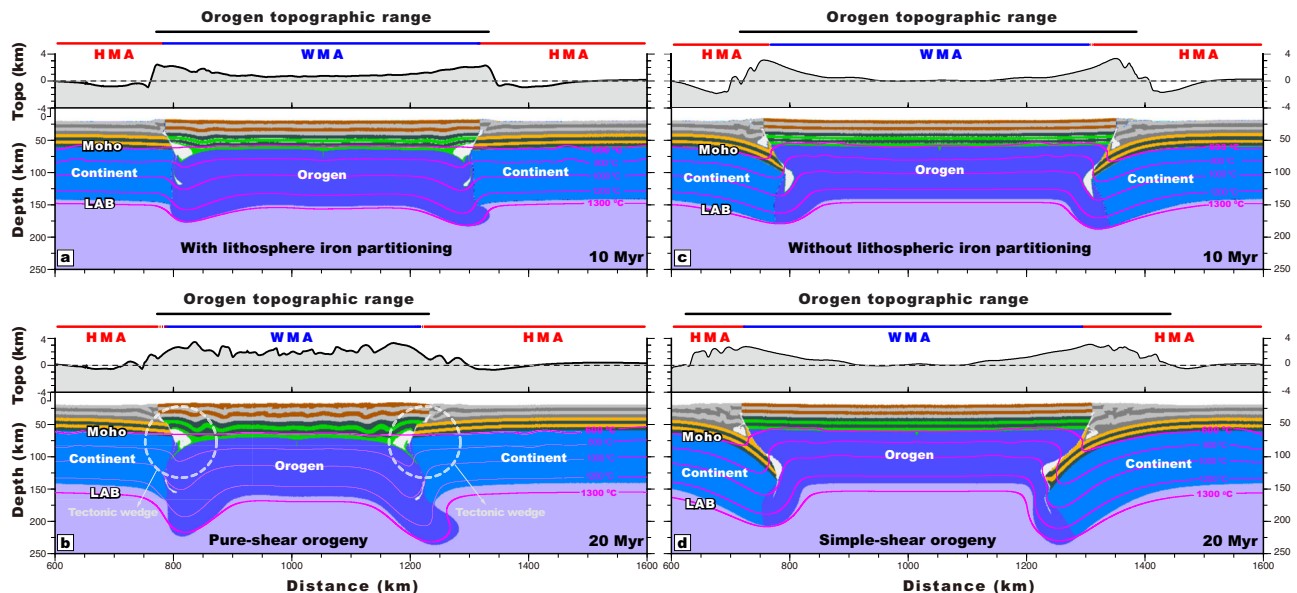

**Fig. 5 | Geodynamic modeling of the role of lower crustal composition and lithospheric mantle depletion on the orogeny style.** Collision between two continents with and without mafic (iron-rich) lower crust and depleted (iron-poor) mantle lithosphere under incoming continents, leading to respectively pure-shear-style thickening of an orogen via tectonic wedging (**a**, **b**) and underthrusting/ subduction of continental lithosphere under the orogen (**c**, **d**). The simulation results are presented at times of 10 Myr (**a**, **c**) and 20 Myr (**b**, **d**) after model initiation. Convergence rate is 10 mm/yr. Parameters of density and viscous rheology contrast between the orogen and its bounding continents are presented in Supplementary Table S1.

## Numerical tests for the hypothesis of Tian Shan pure-shear orogeny

To test our hypothesis, we performed a series of numerical simulations of mountain building in an intracontinental setting ("Methods"). In our reference model, a weaker continental block that represents the future Tian Shan orogen is bounded between two strong continental blocks, representing the Tarim and Kazakhstan cratonic continents (Supplementary Fig. S13 and Supplementary Table S1). A convergence rate is imposed at the right side of the model to account for the far-field effects of the India-Asia collision. A series of numerical experiments that served as sensitivity tests (Supplementary Table S2), together with the geologic-magnetic observations (Figs. 2, 3 and 4), demonstrate that the lithosphere mantle density variation and associated crustal rheology reflected by spatial iron content variations are two critical factors controlling the style of orogenic deformation (Supplementary Figs. S14, S15). When the bounding continents have a mafic lower crust and an iron-depleted underlying lithospheric mantle (depletion-induced density deficit = 30 kg/m³), horizontal shortening is widely distributed across the orogen, forming a bivergent thrust belt with a thick orogenic mantle root that is bounded by a lithosphere-scale tectonic wedge with sharp Moho offsets (Fig. 5a, b). We refer to this style of deformation as a pure-shear orogeny end-member. In this case, the continental crust negligibly projects beneath the orogen after 20 Myr of convergence (< 30 km), and therefore any continental magnetic signal would not be observed within the orogen's interior. These model results, including structural architecture and distributed deformation, are thus similar to geologic-geodetic observations from the Tian Shan[29,31] (Fig. 2).

Conversely, when there are no differences in lower crustal rheology and lithospheric mantle density between the orogen and its bounding continents, deformation involves underthrusting/subduction of the converging continents beneath the orogenic lithosphere such that there is significant overlap (> 100 km) between the converging continental crust and the orogen (Fig. 5c, d). Crustal thickening and uplift concentrate at the collisional-margin zones, and the central orogenic crust is only marginally deformed and thickened (Fig. 5c, d).

We refer to this style of modeled deformation as representing the simple-shear orogeny end-member. These simulations demonstrate that the underthrusting lower crust will remain cooler than the Curie isotherm of 600 °C such that a highly positive magnetic signal would be preserved and observed (Figs. 1, 2, and 3). In this case, the magnetic signal of the converging continents would project into the modeled orogen by > 100 km, similar to the positive long-wavelength magnetic signals projecting into the Canadian Rocky thrust belt in North America[36] (Supplementary Fig. S8a) and the Chinese Daba Shan[45] (Supplementary Fig. S8b).

## Intracontinental orogenic mode and its significance for cratonic lithosphere evolution

Our models demonstrate that iron distribution within the continental crust revealed by aeromagnetic imaging may indeed be indicative of its composition and rheological properties, which control the style of intracontinental orogenic deformation. Although our numerical results suggest that the iron-rich mafic lower crust rheologically coupled to depleted mantle lithosphere resists intra-continental subduction, there may be some notable exceptions to this case. Strong positively buoyant cratons underthrust and subduct beneath orogens when attached to subducting oceanic lithosphere[46,47], such as for the Jurassic Daba Shan[45] and Cenozoic Himalayan[48] orogens. In these examples, preceding oceanic subduction, driven by the negative buoyancy of the oceanic lithosphere compared to the asthenosphere[49], can pull the attached continental crust to depth. Buoyant continental mantle lithosphere should resist continental subduction[46,47], but if the passive margin is stretched/thinned or the low-density upper-middle crust tectonically accretes to the upper plate during convergence, the mafic lower crust may be underthrust and eclogitized[50], as suggested for the Himalaya and Daba Shan[45,48]. The negative buoyancy of the eclogitized lower crust relative to the asthenosphere can overwhelm the positive buoyancy of the melt-depleted cratonic lithosphere. This will operate as a positive feedback, where the negative buoyancy can drive continued plate convergence, continental underthrusting, and subduction and/or delamination[45,48,51].

Conversely, if slab pull force from the attached oceanic lithosphere or eclogitized lower crust is interrupted, say by slab tear[52] or plume impact[53], continued convergence and subduction of the buoyant lithosphere may cease.

Thermal state may also impact orogeny[2,54]. It has been hypothesized that hot upper plate rocks associated with an active arc may be less viscous to promote simple-shear deformation[55]. The Himalaya and Daba Shan examples involved active or recently active arcs, and the warm lithosphere may have weakened the orogen to allow subduction. The southern Canadian Rocky Mountains similarly involve the North American craton subducting westward beneath an active arc system[36]. Conversely, the Cenozoic Tian Shan did not involve a recently active arc, and thus, they may have had a cold orogenic core able to viscously resist subduction. However, in the Andes, both pure- and simple-shear deformation modes are observed along the same arc system[55], which demonstrates that a warm arc does not uniquely drive continental subduction. To further explore this, we conducted ancillary numerical simulations that suggest that a hot orogen may undergo pure-shear deformation due to distributed deformation of the less viscous crust (Supplementary Fig. S16a, S16b), whereas a colder orogen experiences simple-shear deformation as the converging continent is underthrust beneath the cold, more rigid orogen (Supplementary Fig. S16c, S16d). These modeling results are consistent with the interpretation that most Precambrian accretionary orogens, associated with an overall hotter mantle and thermal state, deformed via pure-shear convergence[56]. Considering that the Tian Shan orogen is interpreted herein to have deformed via pure-shear deformation and was not a recently active hot arc, our results suggest that thermal state is not the dominant control on the style of orogeny. Instead, we interpret that lithospheric mantle and crustal compositional variations often more strongly control orogenic mode than the variability of the lithospheric thermal structure.

The influence of lithospheric compositional variability on orogeny style is also important for global plate tectonics and strain localization[21,57]. Pure-shear convergence of buoyant and rigid cratons focuses strain within the rheologically weaker bounding orogens, such as the Tibetan Plateau[58,59], and thus cratons are spared from simple-shear subduction where the continental lithosphere could be subducted due to eclogitization and slab pull[48,59]. This continental pure-shear deformation process explains the longevity and survival of cratonic mantle lithosphere in Earth history despite repeated orogenic cycles[18,60]. Furthermore, most cratons globally are composite cratons formed during Archean-Proterozoic accretionary orogeny associated with homogeneous horizontal shortening and vertical strains, without subduction-related lithospheric mass loss[60]. Hotter mantle potential temperatures in the Precambrian were associated with more rigorous mantle convection and intense plume activity[61,62], which would have led to higher degrees of mantle melting and associated iron depletion[18,63]. Our results suggest that a more melt-depleted mantle lithosphere and hotter orogeny in the Precambrian would favor pure-shear accretionary orogeny, which is consistent with observations from the Archean-Proterozoic cratons[56]. This style of pure-shear orogeny is consistent with the growth of Precambrian cratons through lateral accretion[60] and the long-term survival of cratons through Earth history by avoiding simple-shear continental subduction. The coupled impacts of lower crustal composition and mantle lithosphere depletion therefore strongly control continental interaction, growth and cratonization.

## Methods
### Aeromagnetic data
A regional total-field magnetic map of the Tian Shan and surrounding regions was constructed to examine the crustal magnetic character. Aeromagnetic data within and around the Tian Shan were previously acquired to aid petroleum and mineral exploration before the 2000s by the China Aero Geophysical Survey and Remote Sensing Center for Natural Resources (AGRS) (Supplementary Fig. S1), China Geological Survey. The aeromagnetic compilation combines more than 20 aeromagnetic surveys and a total of ~ 0.67 million-line-km, flown between the 1960s and 2000s with variable flight-line spacings (Supplementary Fig. S1). The original data were processed following previously reported protocols at the China Geological Survey-AGRS (Beijing)[64,65]. To fill the aeromagnetic data gaps, the aeromagnetic data set (5-km grid and 1-km altitude) covering the China mainland, offshore and adjacent areas (CMOA) was merged into the AGRS aeromagnetic data[66,67] (Supplementary Fig. S1). To be consistent with the CMOA dataset, the AGRS dataset was first processed by upward-continuation to 1 km and then resampled at 5 km grid, which was all integrated into the Geosoft Oasis Montaj software package. The total-field magnetic anomalies within and around the Tian Shan were finally compiled with 5-km resolution at 1 km altitude. Finally, the AGRS and CMOA reprocessed aeromagnetic datasets were stitched together (Supplementary Fig. S2) and processed via the differential reduction to the pole method[68] to accurately relocate magnetic source locations and boundaries of the Tian Shan and its environs (Supplementary Fig. S3).

### Long-wavelength magnetic anomaly analysis and crustal iron content estimation
It is thought that lower crustal mafic granulite is sufficiently magnetized to account for amplitudes of long-wavelength magnetic anomalies, corresponding to high-grade metamorphic rocks of deep-seated origin[22,69]. The major magnetic carriers are magnetite, which is a source of long-wavelength magnetic anomalies[22,69]. The amplitude and spatial patterns of magnetic anomalies are primarily caused by variations in the amount, type, and distribution of magnetic minerals, mainly magnetite ($Fe_3O_4$) with a Curie isotherm of 580 °C and its solid solutions ($FeTiO_3$, $Fe_2TiO_4$) in crustal rocks[22,69,70] (Supplementary Fig. S4a). The minimum and maximum magnetic susceptibilities from the China mainland dataset (248,756 field rock samples)[71] (Supplementary Fig. S4b) correspond to two end-member rock types for the lower crust: pyroxene-dominated (iron-rich) and plagioclase-dominated (iron-poor) rocks. We assume the long-wavelength magnetic highs represent the location of mafic iron-rich domains in the lower crust (Supplementary Fig. S4).

To verify whether the long-wavelength magnetic highs reflect iron-rich mafic zones embedded within the lower crust, we used the upward continuation method to filter deeper versus shallow signals at depths of 5, 10, 20, 30, and 40 km (Supplementary Fig. S5). The positive long-wavelength magnetic signals within the Tian Shan-bounding blocks remain stable, whereas the short-wavelength magnetic highs within the Tian Shan orogens become gradually dampened. Constrained by the 15 km thickness of sedimentary rock sequences in the upper crust[72,73], the thickness of the magnetic layer in the crust could be > 25 km, assuming that the lower bound of the crust-scale magnetic zone is > 40 km in thickness. We further employ the three-dimensional (3D) regularization inversion[74] to image the crustal structure of induced magnetization intensity (M) Supplementary Fig. S6, ignoring remnant magnetism. The model consists of $271 \times 121 \times 60$ grid cells at a resolution of $5 km \times 5 km \times 1 km$. The lower crust of the Tarim and Kazakhstan blocks are strikingly expressed by regional high-value magnetic zones, which are the origin of these long-wavelength magnetic highs, while that of the Tian Shan presents no magnetism (Supplementary Figs. S6, S7).

In lower crustal granulite-grade rocks, bulk iron content may indicate the degree of melting of the underlying lithospheric mantle[18] because enriched iron in the lower crust may reflect depletion and extraction from the underlying mantle lithosphere during melting. To approximately quantify the crust-scale iron content, we used the magnetite content in the crust as an indicator because iron-rich rocks have a higher magnetite-producing potential than iron-poor rocks in

the lower crust granulite-grade rocks[22,69,70]. To a very good approximation, the induced magnetization is directly proportional to magnetite content. Magnetite content was calculated using an empirical correlation between the volume magnetic susceptibility ($k$) and the volume percentage of magnetite (V) in the rock, $k \approx 0.033V$[22]. The volume magnetic susceptibility ($k$) is calculated from $k = M\mu_0/\Delta T$[75], where M is the induced magnetization intensity, $\mu_0$ is the magnetic permeability of free space ($\mu_0 = 4\pi \times 10^{-7}$ H/m), and $\Delta T$ is the total geomagnetic field intensity of Tian Shan and its surrounding domains ($\Delta T = 55000$ nT).

## Thermo-mechanical modeling

We use the thermomechanical code I2VIS[76]. The governing equations of conservation of momentum, mass and energy are solved by using a conservative finite difference method and a marker-in-cell technique applied on a staggered non-uniform Eulerian grid. It simulates creeping flow due to thermal and chemical buoyancy forces, and accounts for the effects of adiabatic, shear, latent and radioactive heating. The initial model is 2200 km wide and 400 km deep, and is resolved with a uniform $201 \times 401$ rectangular grid. Over 3.6 million Lagrangian markers are randomly put in the model domain, which are used to advect material properties defined at the Eulerian mesh. The deformation of all materials is governed by visco-plastic rheology, which take account of plastic yielding at shallow depths and low temperatures, and dislocation creep at greater depths and temperatures. The details of the method can be found in the reference[76].

The initial model is composed of three continental blocks: an orogen in the middle and two cratonic continents at both sides (Supplementary Fig. S13). Both the continents and orogen initially have a 40 km-thick crust (the upper and lower crust are 20 km thick) and an 80 km-thick lithospheric mantle, giving a total lithospheric thickness of 120 km. One small seed is imposed at the bottom of the orogen crust, which is used to induce deformation localization at the center of the model. According to the peneplanation of Tian Shan during the Mesozoic tectonic quiescence[26], the initial pre-Cenozoic Moho interface beneath the Tian Shan could be approximately flat.

Two crustal-scale weak zones are placed at the boundaries between the continent and orogen, which are used to facilitate the continental underthrusting beneath the orogen. For the continents, the initial temperature increases linearly from 0 °C at the model surface to 600 °C at the crust base with a geothermal gradient of 15 °C/km, and continues to increase to 1330 °C at the lithosphere base with a geothermal gradient of ~ 9.1 °C/km. For the orogen, the temperature at the Moho is 100 °C hotter than that of the craton, giving a geothermal gradient of 17.5 °C/km (Supplementary Fig. S13). An adiabatic gradient of 0.5 °C/km is used for the sub-lithosphere mantle. All mechanical boundary conditions are free slip, except for the lower boundary, where an external free slip boundary condition is applied[77]. The thermal boundary conditions are constant temperature (273 K) on the top, remote fixed temperature on the bottom[77], and insulating (no horizontal heat flow) on both sides. The surface of the rocky portion is treated as an internal free surface by placing an overlying 20 km-thick 'sticky air'[78], which is characterized by low viscosity ($10^{18} Pa \cdot s$) and low density of $1 kg \cdot m^{-3}$. A convergence rate of 10 mm/yr is imposed at the right side to drive the orogeny (Fig. 4 and Supplementary Fig. S13).

A laboratory-based viscoplastic rheology is used for all the materials. Plastic deformation follows a pressure-dependent Drucker-Prager yield criterion[79]:

$$\sigma_{yield} = C + P \sin\left(\varphi_{eff}\right)$$

$$\eta_{plastic} = \frac{\sigma_{yield}}{2\dot{\varepsilon}_{II}}$$

where $\sigma_{yield}$ is the yield stress; $P$ is the pressure; $C$ is the cohesion at $P = 0$; $\varphi_{eff}$ is the effective internal frictional angle that integrates the effects of internal frictional angle and pore fluid coefficient; $\dot{\varepsilon}_{II} = \left(0.5\dot{\varepsilon}_{ij}\dot{\varepsilon}_{ij}\right)^{1/2}$ is the second invariant of the strain rate tensor $\dot{\varepsilon}_{ij}$; $\eta_{plastic}$ is the viscosity for plastic rheology.

The viscosity for dislocation creep depends on strain rate, pressure and temperature, and can be expressed in terms of deformation invariants as follows[79]:

$$\eta_{creep} = f A_D^{-1/n} \dot{\varepsilon}_{II}^{(1-n)/n} \exp\left(\frac{E_a + PV}{nRT}\right)$$

where $R$ is the universal gas constant, $T$ is the absolute temperature, and $f$ is a scaling factor[80]. The rheological parameters, $A_D$, $n$, $E_a$ and $V$, are the material constant, stress exponent, activation energy and activation volume, respectively.

The effective viscosity $\eta_{eff}$ is given by:

$$\eta_{eff} = \min\left\{\eta_{plastic}, \eta_{creep}\right\}$$

The laboratory-derived flow laws of 'wet quartzite' and 'felsic granulite'[79] are used for the orogen upper and lower crust, respectively. A scaled 'dry olivine' flow law (DOL × 0.25) is used for the orogen mantle[80], and a 'wet olivine' flow law (WOL × 2) for the sublithospheric mantle[79]. The flow law for the craton mantle is a 'dry olivine' for an iron-depletion case, and a scaled 'dry olivine' (DOL × 0.25) for an iron-rich case. We use 'wet quartzite' and zero friction angle to represent crustal weak zones and seeds. The rheological parameters used in the study are presented in Supplementary Table S1.

All materials have a temperature- and pressure-dependent density:

$$\rho = \rho_0 \left[1 - \alpha\left(T - T_0\right)\right]\left[1 + \beta\left(P - P_0\right)\right]$$

where $\rho_0$ is the reference density at room condition ($P_0 = 0.1$ MPa and $T_0 = 298$ K), and $\alpha = 3 \times 10^{-5}$ K$^{-1}$ and $\beta = 1 \times 10^{-5}$ MPa$^{-1}$ are the coefficients of thermal expansion and compressibility, respectively. We employ depletion density to capture the effect of Fe depletion on the density of the lithospheric mantle (i.e., Mg# = 100 × Mg/(Mg + Fe))[18]. The depletion density with respect to the orogen lithospheric mantle measures the chemical depletion of the continent lithospheric mantle due to melt extraction, such as when the cratonic continent lithosphere was generated from Early-Earth higher mantle potential temperature[81] and impinged by a hot mantle plume[82], like the Permian plume-lithosphere interaction occurred in the Tarim Basin[63,64]. Here, it is expressed as $\Delta\rho = \rho_{OLM} - \rho_{CLM}$, where $\rho_{OLM}$ and $\rho_{CLM}$ represent the reference density of the orogen and continent lithospheric mantle, respectively. The $\Delta\rho$ values range from $-60$ to $+60$ kg/m³, which are consistent with variations in lithospheric mantle peridotite density spanning −2.0% and +2.0%[60,83]. The more positive $\Delta\rho$ values correspond to greater percentages of partial melting that a continent lithospheric mantle experienced, whereas a negative $\Delta\rho$ value geologically corresponds to metasomatism of the continent lithospheric mantle peridotite. The reference density for the orogenic lithospheric mantle is fixed at 3300 kg/m³.

To assess the sensitivity of lithospheric mantle depletion density and lower crustal rheology with mafic/felsic compositions on mountain building, we conducted 20 additional model tests (Supplementary Table S2). In these models, the initial thermal structure of the bounding blocks is kept the same as the orogen. The depletion of the mantle lithosphere exerts a strong control on the mode of collisional deformation (Supplementary Fig. S14). A more mafic lower crust generates more simple-shear deformation at the collision zones (Supplementary Fig. S15a), whereas a more felsic lower crust leads to

more limited underthrusting as pure-shear deformation (Supplementary Fig. S15b). Moreover, the thermal state of the orogenic lithosphere is also an important factor controlling the style of orogenic crustal deformation (Supplementary Fig. S16). In contrast to the pure-shear mode in the hot orogen model (Supplementary Fig. S16a, S16b), the model without lateral temperature variations behaves as a simple-shear mode, with the continental lower crust deeply underthrusting/subducting under the orogen (Supplementary Fig. S16c, S16d).

Based on parameter sensitivity analysis, integrated with the multi-method geological-geophysical data, we use the numerical simulations to test the validity and viability of our hypothesis that the lithosphere mantle depletion and associated crustal composition reflected by spatial variations in the Fe content are the two critical factors controlling the style of orogenic deformation.

## Data availability
All the data are included in the article and/or Supplementary Material. The aeromagnetic data covering the Tian Shan and its environs are available in the Source Data file. Source data are provided in this paper.

## Code availability
The thermomechanical code I2VIS is available from Taras Gerya (taras.gerya@eaps.ethz.ch) upon request.

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

## Acknowledgements

We dedicate this manuscript to the memory of Professor An Yin, who passed away unexpectedly while doing what he loved: teaching field

geology to students at UCLA. An's unending enthusiasm, boundless creativity, and positive influence on his friends and colleagues were remarkable. We are greatly grateful for the discussion with Dr. Sean Willet and Dr. Behr Whitney at ETH-Zürich, relocated seismicity dataset support from Dr. Shanshan Liang at China Earthquake Networks Center. This study was supported by the National Natural Science Foundation of China (42422402, U2344203, 42072233 and 41902202), the U.S. National Science Foundation (EAR-1914501), the Second Tibetan Plateau Scientific Expedition and Research (2019QZKK00708), and the Chinese Postdoctoral Science Foundation (2019M652062).

## Author contributions

X.X. designed research; X.X., A.Y., A.V.Z., T.G., L.C., H.L.C., and B.D.W. performed research; X.X., B.D.W., X.T.K., Y.Y.S., and S.H. contributed research data; T.G. provided the thermomechanical code I2VIS; L.C. conducted numerical modeling; X.X., A.Y., T.G., X.H.S., L.W., J.G.L., Y.Y.S., X.B.L., and S.F.Y. analyzed data; X.X., A.V.Z., A.Y., T.G., L.C., and H.L.C. drafted and wrote the paper; All authors reviewed and edited the paper.

## Competing interests

The authors declare no competing interests.

## Additional information

[1]College of Geophysics and Petroleum Resources, and Key Laboratory of Exploration Technologies for Oil and Gas Resources (Ministry of Education), Yangtze University, Wuhan, China. [2]China Aero Geophysical Survey and Remote Sensing Center for Natural Resources, China Geological Survey, Beijing, China. [3]Department of Earth, Planetary, and Space Sciences, University of California-Los Angeles, Los Angeles, California, USA. [4]Departure of Earth Sciences, ETH-Zürich, Zürich, Switzerland. [5]School of Earth Sciences, Zhejiang University, Hangzhou, China. [6]Nevada Bureau of Mines and Geology, Nevada Geosciences, University of Nevada, Reno, Nevada, USA. [7]State Key Laboratory of Lithospheric and Environmental Coevolution, Institute of Geology and Geophysics, Chinese Academy of Sciences, Beijing, China. [8]State Key Laboratory of Geological Processes and Mineral Resources, and Frontiers Science Center for Deep-time Digital Earth, China University of Geoscience (Beijing), Beijing, China. ✉e-mail: xuxigeo@gmail.com; hlchen@zju.edu.cn

