## [Transparent Peer Review file · Nature Communications]

Mode of intracontinental mountain building controlled by lower crustal composition and mantle lithosphere depletion

Corresponding Author: Professor Xi Xu

A version of this paper was originally rejected for publication by Nature Communications, however that decision was reconsidered after appeal by the authors.

Version 0:

Reviewer comments:

Reviewer #1

(Remarks to the Author)

This study tries to investigate mode of intracontinental mountain building by using Tianshan orogenic belt as an example. However, my opinion seems to be not supportive. Because there are so many critical issues should be addressed.

1. One of the biggest problems in this paper is that there are already a lot of deep reflection seismic data including the book of Xiao Y.C., Liu, X., and Gao, R. et al. (2004) about the Tianshan orogenic belt, which can clearly reveal the orogenic pattern, but the authors seem to ignore it.
2. The other major issue with the paper is regarding the interpretation of aeromagnetic data, which happens to be a crucial piece of evidence supporting the final conclusions of this study. The authors attempt to use the relationship between aeromagnetic data and Fe-bearing minerals in the crust to explain the tectonic mechanism. However, based on my long-term experience and deep understanding, such a relationship is evidently questionable. This is because aeromagnetic anomalies that could reflect such large-scale, deep-seated (crustal-scale) mineral magnetic anomalies would necessarily be low-intensity long-wavelength components, which are not demonstrated or reflected in the data processing presented in the paper. Regional aeromagnetic anomalies typically reflect regional metamorphic basement (including metamorphic magnetite, etc.), which is a basic global pattern. From a geophysical perspective, it is recommended to use gravity data for applications that aim to reflect large-scale orogenic styles because the long-wavelength anomalies in Bouguer gravity data reflect changes at the crust-mantle boundary, which may explain orogenic events more effectively than aeromagnetic data.
3. The authors tried to validate the orogenic model using numerical simulation methods, but a major flaw is that the results of numerical simulations are highly dependent on initial and boundary conditions. However, this study fails to analyze the parameter sensitivity of these simulation results. This lack of analysis means that the conclusions drawn may not necessarily be universally applicable or stable, but rather specific to certain conditions or scenarios.

Reviewer #2

(Remarks to the Author)

Review:

Manuscript titled "Mode of intracontinental mountain building controlled by lower crustal composition and mantle lithosphere depletion" by Xu et al. submitted to the Nat. Comms proposes and demonstrates that the iron content in the continental crust and lithospheric mantle, could control the style of continental collisions, pure shear type-distributed shortening, (e.g., Tien Shan) vs discrete shortening involving subduction of the continental lithosphere-simple shear, (e.g Tibet). From the aeromagnetic data they infer that the long-wavelength positive anomalies in the Tarim and Kazakh cratons are localized in the mid-lower crust whereas the short wavelength signal/negative signal in the Tian Shan Orogen corresponds to the upper-mid orogenic crust. They argue that the iron content via its dependence on the buoyancy and strength, which they report to be higher in the foreland/hinterland side lithosphere compared to the one in the orogenic lithosphere in the Tian Shan could have allowed for more distributed deformation upon convergence from the India-Eurasia collision. The authors further extended this observation with an understanding that the cratonic lithospheric mantle is depleted in Fe and the orogenic

lithospheric mantle (Paleozoic-Mesozoic) is enriched with the Fe. Further, using representative models of the continental collision using the thermomechanical models they demonstrate the feasibility of their hypothesis. The authors conclude that the depleted lithospheric mantle of the bounding cratons resists underthrusting/subduction leading to pure shear/distributed deformation in the Tian Shan.

I find the manuscript written well, and the message is clear. Overall, I enjoyed reading it. I have to admit that I am not an expert in magnetic field data inversion, which is primary evidence for their conclusions, hence my review is based on the implications for continental lithosphere deformation. Below are some suggestions which could improve/help the authors to further strengthen their arguments and findings.

Major Points:

Reported thermomechanical models (two), inherently includes a weaker (Hotter orogenic lithosphere), which will lead to distributed deformation on imposed convergence. As authors argue that since the orogenic crust in the Tian Shan is more felsic hence would be hotter than the lithospheres of the bounding cratons making is weaker and leading to the distributed deformation. This clearly hints towards the thermal state leading to weaker orogenic lithosphere to allow for the distributed deformation. However, in the second last paragraph of the manuscript the authors argue that the thermal state of the upper plate is not a decisive factor for the mode of continental deformation. Maybe a model-set without the higher temperatures in the orogenic lithosphere would further strengthen the inference that the lithospheric mantle and crustal compositional variations strongly control orogenic mode (Pure shear vs simple shear).

Decreasing the density of the lithospheric mantle and increasing it for the crust (density?) of the bounding cratons results in net positive buoyancy with respect to the orogenic lithosphere mantle which is taken to be fertile, leads to no subduction of the cratonic lithosphere. It would be interesting to demonstrate which one is more dominating here, the lithosphere mantle (which is thicker) or the continental crust. Maybe a set of models where crust is fixed to a same thermochemical-mechanical properties in both the orogen and foreland/hinterland crust and varied mantle lithospheric mantle composition, would help to further strengthen the conclusion that the composition of the lower crust as well as the lithospheric mantle is a decisive factor. Could the other physical properties (densities in the crust, thermal conductivity and radiogenic heat production) used in the thermomechanical model be included in the SI?

Other comments:

Line 73-75 "Fortunately, iron enrichment in the crust can be detected by aeromagnetic anomalies, and the associated iron distribution in the cratonic lithosphere can be quantified using inverse methods²¹"

It is understood that in the crust it is possible to invert for the iron content using aeromagnetic anomalies, but how could it be used for the lithospheric mantle which could be above the Curie temperature?

Line 126: referencing Fig 2e after "shortening".

Line 151: "widespread shallow crustal seismicity²³ (Fig. S9a),"

It is difficult to see the shallow crust seismicity from the map, all the earthquakes have same colour. Depth information could be included in this figure in order to support the widespread shallow seismicity (with depth: upper crustal level?). Looking at the depth cross-section of the seismicity, in Figure S11 the seismicity, at least, in the southern front of the Tian Shan is not strictly restricted to the shallow depth <20 km which corroborates the Pure shear deformation rather than simple shear or crustal flow (should be hot enough to flow, not the case here because there are earthquakes). Similarly in the interior of the orogen depth distribution of the earthquakes could further support the distributed deformation as well as the temperatures indirectly to support the inferred magnetic signal.

Line: 202-206: "The spatial correspondence between strong long-wavelength magnetic signals and lack of intra plate deformation supports our interpretation that the magnetic signals may be diagnostic for identifying iron distribution in the lithosphere, with an iron-enriched lower mafic crust coupled with a depleted mantle."

Maybe in the crust and lower crust would be the better wording.

Line 396:398: "A dumped 'dry olivine' flow law ($DOL \times 0.25$) is used for the orogen mantle, and a 'wet olivine' flow law ($WOL \times 2$) for the sublithospheric mantle⁶⁹."

Could you please explain the "dumped dry olivine" with a proper reference for the readers as well to me.

Line 212: "To test our hypothesis, we performed a series of numerical simulations of mountain building in an intracontinental setting (Methods)."

Only two models are shown/reported.

Line 213-214 "In our reference model, a weaker continental strip that represents.."

Do you mean stripe?

Lines: 280-281 "This process explains the longevity and survival of cratonic mantle lithosphere in Earth history despite repeated orogenic cycles"

By process do you mean “the pure shear deformation of the cratons” or their inherent chemical composition?

Figure S10: The panel d. figure is also a common-conversion point image and not sure what joint inversion mean here. The difference with the one in panel c. is because of the gaussian widths.

Figure S11: “Magnitude” Scale of Earthquakes.

Figure S12: Is there a reason to truncate the negative values in the map. What does the thick green and turquoise colour line represent? If nothing then please remove to avoid the sensitive confusion/s.

Figure S13: pink vertical bar?

Version 1:

Reviewer comments:

Reviewer #1

(Remarks to the Author)

I am convinced that this paper has not been well revised. The main research conclusions of this paper remain highly questionable for the following reasons:

1. The author has added citations of related papers such as the book of Xiao Y.C., Liu, X., and Gao, R. et al. (2004) without delving deeply into the limitations of the original model. What is the advancement of the model proposed in previous studies? Which specific errors in past research does it bridge? More importantly, on which characteristics of observational data, particularly deep reflection seismic profiles, has the new model been significantly validated? This question needs to be answered affirmatively. Otherwise, the so-called model is merely a hypothesis without much evidence, which will not garner widespread interest from readers.

2. The author acknowledges the significant role of gravity models in explaining deep structures, particularly the undulations of the Moho surface. However, there is still a lack of effective gravity data and their processed results to directly support the research in this paper. This ambiguity greatly reduces the reliability of the research findings.

3. Moreover, from the results of multiple numerical simulations with changed parameters provided by the author, it seems to further confirm the previous doubts, indicating the uncertainty of the model. Each mode obtained from the numerical simulations is different, and the discrepancies are quite significant.

4. It is suggested that the author continue to conduct in-depth research, using observational data, particularly deep reflection seismic data, as the primary constraint conditions for the structure. Geological, geophysical data, including gravity and magnetic data, should be integrated into a rigorous model framework, preferably a quantifiable one. Such research results would be valuable. Speculations based on certain aspects in this paper are not recommended as they do not solve the problem but rather introduce another possibility, which brings more potentially insignificant issues to the study of the Tianshan orogenic belt.

In all, this version of the manuscript can not be accepted for publication.

Reviewer #2

(Remarks to the Author)

I thank and congratulate the authors. They have addressed all the concerns and have modified the manuscript accordingly.

I have no more concerns and recommend publication in Nat. Comms.

Version 2:

Reviewer comments:

Reviewer #3

(Remarks to the Author)

Peer Review of

Mode of intracontinental mountain building controlled by lower crustal composition and mantle lithosphere depletion

Manuscript #NCOMMS-24-13052A

submitted to Nature Communications

This study presents the results of an aeromagnetic study of the Tian Shan orogen and the bounding Tarim Craton and Kazakh Craton. The data reveals different lithospheric composition between the Tian Shan belt and the surrounding cratons: The former has a more felsic lower crust and a less depleted mantle lithosphere, while the latter possesses a more depleted lithospheric keel and a more mafic lower crust. The authors argue that, as a result of the compositional difference, the strong and buoyant cratons resist internal deformation and subduction. The weak Tian Shan belt, on the other hand, deforms by distributed pure shearing. To test this hypothesis, geodynamic modeling is carried out with varying depletion-induced mantle lithospheric density, lower crustal compositions and thermal conditions. The models show that pure shear deformation mode is favored if the less depleted lithospheric mantle and more felsic lower crust are present in the orogenic belt. Otherwise, the crustal deformation is dominated by simple shearing mode or subduction of the bounding craton. Furthermore, the high temperature thermal state also favors distributed pure shearing deformation mode, although the thermal state per se may only play a minor role in controlling the crustal deformation in the Tian Shan belt.

I find the multidisciplinary approaches the authors have taken to formulate and test their tectonic hypothesis interesting and the implications of the study concerning the crustal deformation are of global significance. I notice that geophysical data have been largely used to propose the tectonic model, though there are still some imperfect results that can be done in the future study. To me, the manuscript is generally well-organized and written and is worthwhile for publication in the journal. I do suggest a few improvements on the chain of logic that leads to the final conclusions. I thereby support publication of the manuscript with minor revisions. Some major comments are as follows:

1. I don't recommend using the term 'Kazakh Craton'. We know that the 'craton' has a clearly geological definition. I would suggest using the term 'Kazakhstan Block' in the text.
2. The numerical modeling results presented in Table S3 have an alternative interpretation. In model names TSM03, TSM07, TSM10, TSM13 and TSM16, the lower crustal flow law for both the orogenic the cratonic lower crust are felsic granulite and they all exhibit pure shear deformation mode. In contrast, in model names TSM04, TSM08, TSM11, TSM14 and TSM17, the lower crustal flow laws are both mafic granulite and they all display simple shear mode. This leads to the alternative interpretation that the deformation mode is strongly controlled by the lower crust rheology rather than the density (or viscosity) contrast caused by the differential depletion between the orogenic and cratonic lithosphere. This contradicts with the statement made in Lines 235-237, in the sense that the orogeny can still undergo pure shearing even when there is "no differences in lower crustal rheology and lithospheric mantle density between the orogen and its bounding continents". We suggest the authors to provide further clarification on this in the supplementary materials if not in the main text.
3. The authors used the $\Delta\rho$ value as a proxy for depletion-induced density variation and tested models with $\Delta\rho$ in -60, -30, 0 and 30 kg/m³. We suggest the authors to provide justification as to why these values are chosen and whether they are geologically reasonable.
4. The authors discuss the impact of thermal state on the crustal deformation behavior from a geodynamic modeling perspective (Figure. S16). Although the authors preclude the thermal state as a major contributing factor to the crustal deformation in the Tian Shan belt (Lines 286- 291), We suggest the authors to briefly touch upon the role of thermal state on Precambrian orogeny (e.g. Chardon et al., 2009) and by doing this, generalize the applicability of the current modeling work to a wider range of geological time.

References

D. Chardon, D. Gapais, and F. Cagnard. Flow of ultra-hot orogens: A view from the Precambrian, clues for the Phanerozoic. *Tectonophysics*, 477(3-4):105–118, Nov. 2009. ISSN 00401951. doi: 10.1016/j.tecto.2009.03.008.

Version 3:

Reviewer comments:

Reviewer #3

(Remarks to the Author)

Peer Review of

Manuscript # NCOMMS-24-13052C

Mode of intracontinental mountain building controlled by lower crustal composition and mantle lithosphere depletion

Comments:

The authors have done a thorough improvement addressing all the review comments from the previous round. The manuscript presents a tightly integrated multidisciplinary study combining aeromagnetic analysis and numerical modeling, with profound implications for crustal deformation modes in both Precambrian and Phanerozoic orogenies. In its present form, the manuscript has been significantly improved in terms of logical clarity, as well as the breadth and depth of discussion. Only a few minor, mostly syntactic, edits are needed, so I am happy to recommend acceptance for publication in *Nature Communications* (NC) after these corrections.

Lines 132-134 Since most of the discussion (and in fact the whole article) is centered around differentiating the pure shear and simple shear modes of deformation for the Tian Shan orogenic belt and the surrounding continental blocks, it would be helpful to introduce these key concepts in the first few paragraphs of the Introduction section.

Line 85 repetition of the word "depleted" . . . presence of a depleted iron-poor depleted mantle lithosphere that is rheologically . . .

Lines 87-88 . . . mantle lithosphere . . . bounds the two sides of the Tian Shan belt, which is the latter characterized by more felsic iron-poor lower crust and less depleted, thinner mantle lithosphere.

Line 98 . . . accelerated deformation in the Miocene and is still being actively shortening shortened today.

Reviewer #1 (Remarks to the Author):

This study tries to investigate mode of intracontinental mountain building by using Tianshan orogenic belt as an example. However, my opinion seems to be not supportive. Because there are so many critical issues should be addressed.

1. One of the biggest problems in this paper is that there are already a lot of deep reflection seismic data including the book of Xiao Y.C., Liu, X., and Gao, R. et al. (2004) about the Tianshan orogenic belt, which can clearly reveal the orogenic pattern, but the authors seem to ignore it.

Reply:

We appreciate the reviewer's comment regarding existing literature and geophysical surveys across the Tian Shan. Here we acknowledge that there have been decades of important high-resolution geophysical investigations across the Tian Shan (Figs. R1a and R1b), including the reference Xiao, X.C., Liu, X., and Gao, R. et al. (2004) provided by this reviewer. This specific reference is a Chinese book entitled “**The crustal structure and tectonic evolution of Southern Xinjiang, China/新疆南部地壳结构和构造演化**” (Xiao X.C, Liu X., and Gao R. et al., 2004/肖序常, 刘训, 高锐等, 2004) . We attach the copy of this book in the NC submission system for the reviewers and editors to further review and evaluate (the Related Manuscript File).

While working on this project, we have reviewed and compiled all available deep reflection seismic profiles across the Tian Shan, including Xiao, X.C. et al. (2004), to synthesize the competing Tian Shan construction models. The stated models, including the northward underthrusting, double underthrusting, and southward underthrusting models (corresponding to the Fig. 1 in our main text), are based on deep geophysical profiles and some geological sections (e.g., Allen et al., 1999 Tectonics; Gao et al., 2013 Lithosphere; Sun et al., 2022 Geology; Zhang et al., 2020 GRL, 2022 Geology; Lei et al., 2007 PEPI; Li et al., 2022 NC). The references we cited in the manuscript are high-quality geological and geophysical sections, such as the deep seismic section across the Tian Shan (Fig. R2) (Li Wei et al., 2022 NC). Conversely, this stated reference (Xiao, X.C., Liu, X., and Gao, R. et al., 2004) is a book that is in only available in Chinese and not as accessible for international scientists. Therefore, while originally writing this manuscript, we felt that it was not the most suitable reference for a paper like this submitted to the international, and broad-readership Nature Communications.

Our original writing and presentation of models for the orogenic architecture included a comprehensive review of all possibilities, including the one presented in the Xiao et al. (2004) book. However, we agree with this reviewer here that this is an important reference, as well as other important papers (e.g., Gao et al., 2013 Lithosphere), and therefore we now cite them in our setup of existing work.

We do fully appreciate that such uncited literature and highlighted comments are important to consider. Based on the P-wave velocity structure on lithospheric scale (Fig. R3a), the authors of this book proposed a double-underthrusting model that Junggar and Tarim lithosphere underthrust beneath the Tian Shan orogen by face to face (Fig. R3b). In the revised manuscript, we have now added two additional citations (Xiao et al., 2004 Book; Gao et al., 2013 Lithosphere), especially related to our Fig.1d in the main text (double-size underthrusting beneath the Tian Shan).

Furthermore, during this revision process we have carefully checked the geophysical data and observational figures from Xiao et al. (2004). During this evaluation process, we note only a few deep reflection profiles (i.e., two seismic sections) in the book. All the geophysical text and figures within the book involve in the pages from Page 2 to Page 53, as well as the supplementary figures at the end. We have pasted this information below, and now better cite this study in the main manuscript. Thank you.

[REDACTED]	[REDACTED]
Fig.R1a Location of deep seismic profiles in the West Kunlun-Tarim-Tian Shan domain	Fig.R1b Crustal structure model in the West Kunlun-Tarim-Tian Shan domain

[REDACTED]	
Fig.R2 Deep seismic section across the Tian Shan (Li Wei et al., 2022 NC)	
[REDACTED]	[REDACTED]
Fig.R3a Tomographic image of P waves across the Junggar-Tian Shan-Tarim domain	Fig.R3b Double-underthrusting model proposed by the authors in this book (Xiao, Liu and Gao et al., 2004)

[REDACTED]	[REDACTED]
[REDACTED]	[REDACTED]
[REDACTED]	[REDACTED]

[REDACTED]	[REDACTED]
---	--

[REDACTED]	[REDACTED]
[REDACTED]	[REDACTED]

[REDACTED]	[REDACTED]
[REDACTED]	[REDACTED]

[REDACTED]	[REDACTED]
---	--

[REDACTED]	[REDACTED]
---	--

[REDACTED]	[REDACTED]
--	---

[REDACTED]	[REDACTED]
[REDACTED]	[REDACTED]

2. The other major issue with the paper is regarding the interpretation of aeromagnetic data, which happens to be a crucial piece of evidence supporting the final conclusions of this study. The authors attempt to use the relationship between aeromagnetic data and Fe-bearing minerals in the crust to explain the tectonic mechanism. However, based on my long-term experience and deep understanding, such a relationship is evidently questionable. This is because aeromagnetic anomalies that could reflect such large-scale, deep-seated (crustal-scale) mineral magnetic anomalies would necessarily be low-intensity long-wavelength components, which are not demonstrated or reflected in the data processing presented in the paper. Regional aeromagnetic anomalies typically reflect regional metamorphic basement (including metamorphic magnetite, etc.), which is a basic global pattern. From a geophysical perspective, it is recommended to use gravity data for applications that aim to reflect large-scale orogenic styles because the long-wavelength anomalies in Bouguer gravity data reflect changes at the crust-mantle boundary, which may explain orogenic events more effectively than aeromagnetic data.

Reply:

We appreciate the reviewer's important comments about the interpretation of aeromagnetic anomaly impacting the modes of mountain building. We agree with reviewer #1's viewpoints that "*.....aeromagnetic anomalies that could reflect such large-scale, deep-seated (crustal-scale) mineral magnetic anomalies would necessarily be low-intensity long-wavelength components.....*". From a geophysical perspective, there are at least two methods that can be applied to delineate magnetic sources at various depths and scales. One method involves potential field conversion, which encompasses derivation and upward continuation. Derivation can emphasize shallow magnetic source (Blackly, 1996), whereas the upward continuation can effectively delineate deeper magnetic source, such as the metamorphic basement and mid-lower crust (e.g., Blackly, 1996; Xu et al., 2023 Geology). The other method is magnetic inversion, which can directly provide the magnetization intensity (magnetic susceptibility) at any depths above the Curie surface (e.g., Hu et al., 2019 Geophysics; Xu et al., 2023 Geology). Magnetic inversion is a well-established and well-proven method that has been widely utilized to delineate the magnetic structure of the whole crust (e.g., Xu et al., 2023 Geology), even the uppermost lithospheric mantle (Ferre et al., 2021 Nature REE).

In our manuscript, we used upward continuation on the magnetic grid data (Fig. S5 in our Supplementary Material file; also pasted below). The magnetic anomalies

processed by upward continuation at various altitude still exhibit low values in the Tian Shan, suggestive of weak magnetic properties of the deep crust in this orogenic belt. Furthermore, the 3D magnetic inversion was also utilized (Fig. S6 in our Supplementary Material file; also pasted below), which revealed that the middle-lower crust of the Tian Shan is characterized by weak magnetization intensity (Fig. S6). Therefore, through these two data-processing methods, we can effectively decipher the information about large-scale, deep-seated “mineral magnetic anomalies” and calculated the Fe content via the empirical correlation between the magnetic susceptibility and the magnetite (Fe₃O₄) percentage (Grant, 1985 Geoexploration).

Fig. S5 Map of magnetic anomalies at different upward continuation values to enhance regional and deep magnetic features. Panels a through f correspond to upward continuation of 1 km, 5 km, 10 km, 20 km, 30 km and 40 km, respectively.

Fig. S6

Fig. S6 Crust-scale magnetization intensity model for the Tian Shan and its bounding cratons.

Here we wish to emphasize that we have also provided details of our processing methods in the Supplementary Material file.

The reviewer #1 noted that “...Regional aeromagnetic anomalies typically reflect regional metamorphic basement (including metamorphic magnetite, etc.), which is a basic global pattern...”. According to classic geomagnetic literature (e.g., Krutikhovskaya and Pashkevich, 1979 *Journal of Geophysics*; Frost and Shive, 1986 *JGR*; Shive, 1989 *GRL*), regional (high-amplitude, long-wavelength) aeromagnetic anomalies reflect contributions from local metamorphic basement but also the middle-lower crust as well as the upper mantle (Friedman et al., 2014 *Tectonophysics*; Ferre et al., 2021 *Nature REE*). Moreover, for example, the 3-D inversion of the aeromagnetic data can reveal the vertical and lateral compositions at the crustal scale (0-40 km).

(Kolawole et al., 2017 GRL), providing key information to decipher the mechanism of great earthquake generation. Therefore, aeromagnetic anomalies have proven to be effective data in imaging the whole crustal structure.

We also concur with reviewer #1's comment: “.....*long-wavelength anomalies in Bouguer gravity data reflect changes at the crust-mantle boundary.....*”. Bouguer gravity data can be used to assess Moho topography, particularly in regions lacking seismic reflection and receiver function survey data. However, there are at least two challenges when interpreting the orogenic structures from such data. First, both lithospheric density heterogeneity and Moho topography can contribute to the observed gravity anomaly (Mooney and Kaban, 2010 JGR; Wang et al., 2014 Tectonophysics). It is impossible to differentiate anomalies originating from lithospheric density heterogeneity and Moho topography if there is no seismic-based Moho constraint (Guo et al., 2019 EPSL). Therefore, determining the Moho from only Bouguer gravity can be very uncertain. Second, even if Moho topography is relatively accurately determined from Bouguer gravity, it is still challenging to determine the mechanism behind Moho topography, as discrete continental subduction, distributed pure-shear shortening, and

ductile flow in the middle and/or lower crust can all influence Moho topography. Therefore, here we did not utilize gravity data to investigate the crust-mantle boundary due to its uncertainty and limited relevance to our research theme. We welcome other studies that attempt such methods, but we did not feel that this would be the best approach to address our research goals. In contrast, magnetic anomalies are sensitive to Fe-bearing minerals, and the presence of Fe-bearing minerals affects lithospheric density distribution and viscous rheology, which are key factors influencing the mechanism of intracontinental mountain building.

We appreciate this discussion to sharpened our text.

Thank you.

- Blakely, R. J. (1996). Potential theory in gravity and magnetic applications. Cambridge university press.
- Frost, B. R., & Shive, P. N. (1986). Magnetic mineralogy of the lower continental crust. *Journal of Geophysical Research: Solid Earth*, 91(B6), 6513-6521.
- Friedman, S. A., Feinberg, J. M., Ferré, E. C., Demory, F., Martín-Hernández, F., Conder, J. A., & Rochette, P. (2014). Craton vs. rift uppermost mantle contributions to magnetic anomalies in the United States interior. *Tectonophysics*, 624, 15-23.
- Ferré, E. C., Kuzenko, I., Martín-Hernández, F., Ravat, D., & Sanchez-Valle, C. (2021). Magnetic sources in the Earth's mantle. *Nature Reviews Earth & Environment*, 2(1), 59-69.
- Grant, F. S. (1985). Aeromagnetism, geology and ore environments, I. Magnetite in igneous, sedimentary and metamorphic rocks: an overview. *Geoscientific Exploration*, 23(3), 303-333.
- Hu, M., Yu, P., Rao, C., Zhao, C., & Zhang, L. (2019). 3D sharp-boundary inversion of potential-field data with an adjustable exponential stabilizing functional. *Geophysics*, 84(4), J1-J15.
- Kolawole, F., Atekwana, E. A., Malloy, S., Stamps, D. S., Grandin, R., Abdelsalam, M. G., ... & Shemang, E. M. (2017). Aeromagnetic, gravity, and Differential Interferometric Synthetic Aperture Radar analyses reveal the causative fault of the 3 April 2017 Mw 6.5 Moiyabana, Botswana, earthquake. *Geophysical Research Letters*, 44(17), 8837-8846.
- Krutikhovskaya, Z. A., & Pashkevich, I. K. (1979). Long-wavelength magnetic anomalies as a source of information about deep crustal structure. *Journal of Geophysics*, 46(1), 301-317.
- Mooney, W. D., & Kaban, M. K. (2010). The North American upper mantle: Density, composition, and evolution. *Journal of Geophysical Research: Solid Earth*, 115(B12).
- Shive, P. N. (1989). Can remanent magnetization in the deep crust contribute to long wavelength magnetic anomalies?. *Geophysical Research Letters*, 16(1), 89-92.
- Xu, X., Chen, H., Zusa, A. V., Yin, A., Yu, P., Lin, X., ... & Wang, B. (2023). Phanerozoic cratonization by plume welding. *Geology*, 51(2), 209-214.

3. The authors tried to validate the orogenic model using numerical simulation methods, but a major flaw is that the results of numerical simulations are highly dependent on initial and boundary conditions. However, this study fails to analyze the parameter sensitivity of these simulation results. This lack of analysis means that the conclusions drawn may not necessarily be universally applicable or stable, but rather specific to certain conditions or scenarios.

Reply:

We appreciate and acknowledge the comments the reviewer brought up here. For these types of numerical simulations, the initial and boundary conditions are critical constraints for how the models will respond to parameter variations. In the past decade, series of numerical and physical modeling tests suggest that the lithospheric-scale rheology and density (lower crust and mantle lithosphere) primarily control the lithosphere deformation and structure in the intracontinental tectonism (e.g., Calignano et al., 2015 *Tectonics*; Chen et al., 2017 *Nature Communications*; Huangfu et al., 2021 *Nature Communications*).

In this work, we first provide multi-method geological-geophysical data that we interpret shows that lower crustal composition and mantle lithosphere depletion (rheology and density) control the mode of intracontinental mountain building. Our primary data revolves around these geological and geophysical datasets. From these datasets, we construct a hypothesis that the partitioning of lithospheric iron can simultaneously influence lithospheric density, rheology, and magnetic structure to influence the orogenic mode. We tested this hypothesis with a series of numerical simulations. Therefore, here we wish to point out that our interpretations are not uniquely dependent on the numerical simulations, but rather the geophysical and geologic observations. The numerical simulations serve to test the validity and viability of our hypothesis.

That said, we appreciate the fact that any numerical simulations are highly dependent on the initial and boundary conditions. The parameter sensitivity analysis is indeed very important pre-condition for numerical modeling and geological interpretation. We have therefore conducted additional parameter sensitivity tests including 20 additional models, which are now presented in the supplementary material (Table S3; Such table screenshot pasted below) and discussed in the revised text. In these models, we keep the initial thermal structure of the bounding cratons the same as

that of the orogen, and test the influences of the depletion density of the craton mantle and the lower crustal rheology with different compositions. The new results show that the depletion-induced density reduction of the cratonic mantle lithosphere is a key factor controlling the style of crustal deformation at the collisional zones (Fig. R4). It can also be seen that the models with mafic lower crust exhibit obvious simple shear behavior (Fig. R5a), whereas the ones with felsic lower crust have limited lower crustal underthrusting, behaving as pure shear (Fig. R5b). The additional numerical experiments together with the original ones presented in the text demonstrate that the lithosphere mantle density and associated crustal composition (and thereby rheology) reflected by spatial variations in the Fe content are two critical factors controlling the style of orogenic deformation.

Table S3

Table S3 Parameters and Results of Conducted Experiments

Model name	Orogen Moho temperature (°C)	Craton Moho temperature (°C)	Orogen lower crust flow law	Craton lower crust flow law	Depletion density ^a (kg/m ³)	Deformation Mode
TSM01	700	600	Felsic granulite	Mafic granulite	30	Pure shear
TSM02	600	600	Felsic granulite	Mafic granulite	30	Simple shear
TSM03	600	600	Felsic granulite	Felsic granulite	30	Pure shear
TSM04	600	600	Mafic granulite	Mafic granulite	30	Simple shear
TSM05	700	600	Felsic granulite	Mafic granulite	0	Simple shear
TSM06	600	600	Felsic granulite	Mafic granulite	0	Simple shear
TSM07	600	600	Felsic granulite	Felsic granulite	0	Pure Shear
TSM08	600	600	Mafic granulite	Mafic granulite	0	Simple shear
TSM09	600	600	Felsic granulite	Mafic granulite	60	Pure Shear
TSM10	600	600	Felsic granulite	Felsic granulite	60	Pure Shear
TSM11	600	600	Mafic granulite	Mafic granulite	60	Simple shear
TSM12	600	600	Felsic granulite	Mafic granulite	-30	Simple shear
TSM13	600	600	Felsic granulite	Felsic granulite	-30	Pure Shear
TSM14	600	600	Mafic granulite	Mafic granulite	-30	Simple shear
TSM15	600	600	Felsic granulite	Mafic granulite	-60	Simple shear
TSM16	600	600	Felsic granulite	Felsic granulite	-60	Pure Shear
TSM17	600	600	Mafic granulite	Mafic granulite	-60	Simple shear

^a A negative depletion density means that the lithospheric mantle of the craton is denser than that of the orogen.

Table S3 in the supplementary material file

Fig. R4 Geodynamic modeling of the role of lithospheric depletion on the orogeny style. Collision between two continents with iron-fertile and iron-depleted mantle density varying from $\Delta\rho=+30$ to -60 kg/m^3 . The models with lithospheric mantle depletion-induced density contrast of -60 kg/m^3 (a) and -30 kg/m^3 (b) lead to pure shear-style thickening of an orogen via tectonic wedging, whereas 0 kg/m^3 (c) and $+30$ kg/m^3 (d) leads to underthrusting / subduction of continental lithosphere under the orogen.

Here, it is expressed as $\Delta\rho_m = \rho_{OLM} - \rho_{CLM}$, where ρ_{OLM} and ρ_{CLM} represent the reference density of the orogen and continent lithospheric mantle, respectively.

The lower crust of the continental and orogenic lithosphere are equipped with mafic and felsic rocks, respectively.

Fig. R5 Geodynamic modeling of the role of lower crustal composition on the orogeny style. Collision between two continents with mafic (a) and felsic (b) orogenic lower crust. Orogen-bounding continents are here equipped with mafic lower crust. The models are equipped with lithospheric mantle depletion density of -30 kg/m^3 .

The addition of these sensitivity tests will be valuable to future readers to assess controls on the styles of orogeny. Thank you for this valuable suggestion. We do note that due to length limitations for the journal, we chose to only display the represent model results in the main text of the manuscript.

Thank you.

Reviewer #2 (Remarks to the Author):

Review:

Manuscript titled “Mode of intracontinental mountain building controlled by lower crustal composition and mantle lithosphere depletion” by Xu et al. submitted to the Nat. Comms proposes and demonstrates that the iron content in the continental crust and lithospheric mantle, could control the style of continental collisions, pure shear type-distributed shortening, (e.g., Tien Shan) vs discrete shortening involving subduction of the continental lithosphere-simple shear, (e.g Tibet). From the aeromagnetic data they infer that the long-wavelength positive anomalies in the Tarim and Kazakh cratons are localized in the mid-lower crust whereas the short wavelength signal/negative signal in the Tian Shan Orogen corresponds to the upper-mid orogenic crust. They argue that the iron content via its dependence on the buoyancy and strength, which they report to be higher in the foreland/hinterland side lithosphere compared to the one in the orogenic lithosphere in the Tian Shan could have allowed for more distributed deformation upon convergence from the India-Eurasia collision. The authors further extended this observation with an understanding that the cratonic lithospheric mantle is depleted in Fe and the orogenic lithospheric mantle (Paleozoic-Mesozoic) is enriched with the Fe. Further, using representative models of the continental collision using the thermomechanical models they demonstrate the feasibility of their hypothesis. The authors conclude that the depleted lithospheric mantle of the bounding cratons resists underthrusting/subduction leading to pure shear/distributed deformation in the Tian Shan.

I find the manuscript written well, and the message is clear. Overall, I enjoyed reading it. I have to admit that I am not an expert in magnetic field data inversion, which is primary evidence for their conclusions, hence my review is based on the implications for continental lithosphere deformation. Below are some suggestions which could improve/help the authors to further strengthen their arguments and findings.

Reply:

We sincerely appreciate and acknowledge the reviewer's constructive review comments.

Major Points:

Reported thermomechanical models (two), inherently includes a weaker (Hotter orogenic lithosphere), which will lead to distributed deformation on imposed convergence. As authors argue that since the orogenic crust in the Tian Shan is more felsic hence would be hotter than the lithospheres of the bounding cratons making it weaker and leading to the distributed deformation. This clearly hints towards the thermal state leading to weaker orogenic lithosphere to allow for the distributed deformation. However, in the second last paragraph of the manuscript the authors argue that the thermal state of the upper plate is not a decisive factor for the mode of continental deformation. May be a model-set without the higher temperatures in the orogenic lithosphere would further strengthen the inference that the lithospheric mantle and crustal compositional variations strongly control orogenic mode (Pure shear vs simple shear).

Reply:

We greatly appreciate the reviewer's comments so that we can better clarify the text and discussion of the impact of thermal state. The thermal state of the orogenic lithosphere is an important factor controlling the style of orogenic crustal deformation. Our original writing noted how examples in the literature are not conclusive for how thermal state predictably impacts orogeny. The cited examples explored deformation involve active or recently active arc systems (e.g., the Andes or Himalaya). For the Tian Shan, there is no geological reason for the Tian Shan crust to be especially hot or cold, given arc activity had ceased by the Triassic.

However, we think the reviewer brings up a valuable comment, and we have conducted additional modeling to explore how thermal variations would impact the mode of orogeny. The hot orogen model results in a pure shear mode of deformation because the orogen is less viscous and deformable (Figs. R6a, R6b) whereas the model without lateral temperature variations deforms via a simple-shear mode, with the cratonic lower crust deeply underthrusting under the orogen (Figs. R6c, R6d). Given our geologic and geophysical interpretations suggest that pure-shear orogeny is active in the Tian Shan, and we have no reason to interpret that the Tian Shan was especially

hot in the Cenozoic, we interpret that thermal state is not the first order control on orogenic mode. Instead, the lithospheric mantle and crustal compositional variations may more strongly control orogenic mode.

We have now added these model results to the Supplemental Materials and have modified the text to make this clear. Thank you for the valuable suggestion, which allowed us to improve our work.

Decreasing the density of the lithospheric mantle and increasing it for the crust (density?) of the bounding cratons results in net positive buoyancy with respect to the orogenic lithosphere mantle which is taken to be fertile, leads to no subduction of the cratonic lithosphere. It would be interesting to demonstrate which one is more dominating here, the lithosphere mantle (which is thicker) or the continental crust. Maybe a set of models where crust is fixed to a same thermochemical-mechanical properties in both the orogen and foreland/hinterland crust and varied mantle lithospheric mantle composition, would help to further strengthen the conclusion that the composition of the lower crust as well as the lithospheric mantle is a decisive factor. Could the other

physical properties (densities in the crust, thermal conductivity and radiogenic heat production) used in the thermomechanical model be included in the SI?

Reply:

We here sincerely acknowledge the reviewer’s important comment and consideration. Our manuscript is an observation-based paper. The logic of our manuscript is that we decipher the crustal compositional structure and propose the geologic hypothesis (Fig.3) via our high-resolution aeromagnetic observation, and then finally test the hypothesis via the numerical modeling. Based on our magnetic observations, we propose that the lower crust composition (rheology) and lithospheric mantle depletion (density) control the mode of lithospheric-scale mountain building. We highlight in our text that the continental crust and lithospheric mantle work together to control the mode, of which the composition is chemically coupled (Fig. 3b and 3c).

As the reviewer pointed out: “... It would be interesting to demonstrate which one is more dominating here, the lithosphere mantle (which is thicker) or the continental crust...”, we agree with this suggestion to run additional numerical models. In one group of models, we keep the physical properties of the orogenic crust and cratonic crust fixed, and systematically vary cratonic lithospheric mantle density (affected by depletion), spanning from 3240 to 3360 kg/m³. In the other group of models, we keep the physical properties of the cratonic lithospheric mantle fixed, and change the flow law of the orogenic or cratonic lower crust (affected by its composition). It can be seen

that as depletion density decreases, the deformation style at the collisional zone changes from pure shear to simple shear (Fig. R4). When the flow law of the orogenic lower crust changes from mafic granulite to felsic granulite, the deformation mode switches from simple shear to pure shear (Fig. R5). These modeling results demonstrate that either a depleted cratonic mantle (Figs. R4a, R4b) or a felsic orogenic lower crust (Fig. R5b) suppress underthrusting of the cratonic lithosphere. Moreover, to our knowledge, there are existing literature that have systematically explored these points, including for example Chen et al. (2017 Nature Communications) and Huangfu et al. (2021 GRL).

Lastly, to the reviewer's comment: "*Could the other physical properties (densities in the crust, thermal conductivity and radiogenic heat production) used in the thermomechanical model be included in the SI?*", we now conducted additional tests with variable physical properties, including crustal density, thermal conductivity and radiogenic heat production, in the SI table.

Accordingly, we have rewritten this portion of the text to address these points. Thank you for the suggestion.

Other comments:

Line 73-75 "Fortunately, iron enrichment in the crust can be detected by aeromagnetic anomalies, and the associated iron distribution in the cratonic lithosphere can be quantified using inverse methods²¹"

It is understood that in the crust it is possible to invert for the iron content using aeromagnetic anomalies, but how could it be used for the lithospheric mantle which could be above the Curie temperature?

Reply:

We do appreciate and acknowledge the comments the reviewer brought up here. Please note the Fig. 3b and 3c in the main text. The crustal scale iron content will indicate the interaction between the crust and its underlying lithospheric mantle (Grant, 1985 Geoexploration). Thus, it is assumed that the iron in the crust is geochemically coupled with the lithospheric mantle. The iron content within the crust will be inversely

proportional to that within the lithospheric mantle. Therefore, we can use the iron content to indicate the state of lithospheric mantle. We expand on this discussion in the main text.

Line 126: referencing Fig 2e after “shortening”.

Reply:

Thanks. It is revised.

Line 151: “widespread shallow crustal seismicity²³ (Fig. S9a),”

It is difficult to see the shallow crust seismicity from the map, all the earthquakes have same colour. Depth information could be included in this figure in order to support the widespread shallow seismicity (with depth: upper crustal level?,). Looking at the depth cross-section of the seismicity, in Figure S11 the seismicity, at least, in the southern front of the Tian Shan is not strictly restricted to the shallow depth <20 km which corroborates the Pure shear deformation rather than simple shear or crustal flow (should be hot enough to flow, not the case here because there are earthquakes). Similarly in the interior of the orogen depth distribution of the earthquakes could further support the distributed deformation as well as the temperatures indirectly to support the inferred magnetic signal.

Reply:

The seismicity data cited in the Fig. S9a is downloaded from the USGS website (<https://www.usgs.gov/>). The earthquake epicenters from the USGS dataset are not all accurately relocated. In contrast, the earthquake epicenters from the China Earthquake Networks Center (CENC) in Fig. S11 are all accurately relocated with some depth uncertainties. Thus, in the Fig. S9a, we only present the distribution of earthquake epicenters. According to the reviewer’s suggestions, we replot the Fig. S9a, presenting the depth information of earthquake epicenter.

Line: 202-206: “The spatial correspondence between strong long-wavelength magnetic signals and lack of intra plate deformation supports our interpretation that the magnetic signals may be diagnostic for identifying iron distribution in the lithosphere, with an iron-enriched lower mafic crust coupled with a depleted mantle.”

Maybe in the crust and lower crust would be the better wording.

Reply:

We have revised the text accordingly. Thank you for the suggestion.

Line 396:398: “A dumped ‘dry olivine’ flow law ($DOL \times 0.25$) is used for the orogen mantle, and a ‘wet olivine’ flow law ($WOL \times 2$) for the sublithospheric mantle⁶⁹.”

Could you please explain the “dumped dry olivine” with a proper reference for the readers as well to me.

Reply:

We greatly appreciate this reviewer’s constructive comments and directional suggestions.

“Dumped dry olivine” means that the laboratory-derived flow law is multiplied

by a scaling factor f . This is because the extrapolation of laboratory data to geological conditions is subject to considerable uncertainty. The scaling approach allows for an assessment of model sensitivity to reasonable variations in viscous strength without invoking additional flow laws, each with their own uncertainties (Beaumont et al., 2006).

In order to keep it simple and address the reviewer's comment here, we renamed it to be "scaled" dry olivine. In a word, it is just an olivine flow law that is scaled by a factor (Beaumont et al., 2006). We further revise the "dumped dry olivine" to be "scaled dry olivine" in the main text and add a proper reference for it.

Beaumont, C., Nguyen, M. H., Jamieson, R. A., & Ellis, S. (2006). Crustal flow modes in large hot orogens (Vol. 268, pp. 91–145). Geological Society Special Publication. <https://doi.org/10.1144/GSL.SP.1>.

Line 212: "To test our hypothesis, we performed a series of numerical simulations of mountain building in an intracontinental setting (Methods)."

Only two models are shown/reported.

Reply:

Thanks for the reviewer's important consideration and suggestion. Actually, we have tested and conducted a series of numerical experiments for mountain building in the intracontinental setting (Please see the updated supplementary material). Considering this is a primarily observation-based study, we only put two numerical models in the main text. During the revision, the parameter sensitivity study has been extended by running 20 additional models, which are now presented in the supplementary material and discussed in the revised paper.

Line 213-214 "In our reference model, a weaker continental strip that represents.."

Do you mean stripe?

Reply:

In this case, we meant a continental checked strip, like a ribbon of continent / block. We have modified the text here to be more clear for future readers. Thank you.

Lines: 280-281 “This process explains the longevity and survival of cratonic mantle lithosphere in Earth history despite repeated orogenic cycles”

By process do you mean “the pure shear deformation of the cratons” or their inherent chemical composition?

Reply:

Thanks. Here we mean the former. Pure shear deformation of cratonic lithosphere will maintain and strengthen its longevity and survival, as compared with simple-shear underthrusting and subduction of continental lithosphere that will lead to its recycling in the mantle. In light of this comment, we have further clarified the text here.

Figure S10: The panel d. figure is also a common-conversion point image and not sure what joint inversion mean here. The difference with the one in panel c. is because of the gaussian widths.

Reply:

Thanks for the reviewer’s important consideration and valuable comments.

The panels c and d are cited from the Zhang et al. (2020 GRL) and Li et al. (2021 NC). Zhang et al. and Li et al. used same observed data and process it via different methods. Here we just cited these processed results to reinterpret the crustal deformation. More details about the data-processing methods will be traced in Zhang et al. (2020 GRL) and Li et al. (2021 NC).

Figure S11: “Magnitude” Scale of Earthquakes.

Reply:

Thanks for the reviewer’s sincere reminder. It is a typo. We revise it here.

Figure S12: Is there a reason to truncate the negative values in the map. What does the thick green and turquoise colour line represent? If nothing then please remove to avoid the sensitive confusion/s.

Reply:

Thanks for the reviewer's important consideration. Here, we want to emphasize the long-wavelength, high-amplitude, positive magnetic anomalies, which directly indicate the domain of iron-rich lower crust. The thick green colour line represents the boundary of these sedimentary basins with cratonic lithosphere, including Tarim, Junggar, Qaidam, Ordos, and Sichuan. The turquoise line represents the boundary of our high-resolution aeromagnetic data. As the reviewer highlighted, we add the label expression in the caption.

Figure S13: pink vertical bar?

Reply:

Thanks for the reviewer's sincere reminder. It is a typo, and should be black vertical bar. We revise it here. Thanks again.

In the second review bound (NCOMMS-24-13052A)

Reviewer #1 (Remarks to the Author):

I am convinced that this paper has not been well revised. The main research conclusions of this paper remain highly questionable for the following reasons:

1. The author has added citations of related papers such as the book of Xiao X.C., Liu, X., and Gao, R. et al. (2004) without delving deeply into the limitations of the original model. What is the advancement of the model proposed in previous studies? Which specific errors in past research does it bridge? More importantly, on which characteristics of observational data, particularly deep reflection seismic profiles, has the new model been significantly validated? This question needs to be answered affirmatively. Otherwise, the so-called model is merely a hypothesis without much evidence, which will not garner widespread interest from readers.

Reply:

This above referenced book proposes a key model referred to as the two-sided subduction model (Figs. 1, 2, also referenced in our original Fig. 1d). As Reviewer #1 pointed out, the deep reflection seismic profile presented in this book is confined to the interaction domain between the Tarim Basin and West Kunlun range (Fig. 3), and *does not* cover any part of the Tian Shan, which is the focus of our study. In preparing this manuscript, we have reviewed and compiled all available deep reflection seismic profiles across the Tian Shan from the literature, including those from the book edited by Xiao et al. (2004) (Fig. 3), Makarov et al. (2010) and Gao et al. (2013) (Figs. 4, 5).

Fig. 1 Two-sided subduction (Xiao X.C., Liu X., and Gao R. et al., 2004)

Fig. 2. The original Fig. 1d in our rejected manuscript.

[REDACTED]	[REDACTED]
Across the Tian Shan Makarov et al., 2010	In front of the southwestern Tian Shan Gao et al., 2013
Fig. 4 Locations of deep reflection seismic profile near the research area	

We acknowledge that the existing deep reflection seismic profiles from these sources (Makarov et al., 2010; Gao et al., 2013) are limited in spatial coverage and resolution, and therefore not highly valuable for constraining the crustal structure of the entire Tian Shan (Figs. 4, 5). Instead, we rely on high-quality seismic profiles, such as those presented by (e.g., Li et al., 2022 Nat. Commun.) to elucidate the fine-scale crustal structure.

[REDACTED]	[REDACTED]
Tian Shan Makarov et al., 2010	Southwestern Tian Shan Gao et al., 2013
Fig. 5 Interpretation of deep reflection seismic profiles the Reviewer #1 highlighted	

Reviewer #1 raised three key questions: “What is the advancement of the model proposed in previous studies? Which specific errors in past research does it bridge? More importantly, on which characteristics of observational data, particularly deep reflection seismic profiles, has the new model been significantly validated?”. We address these questions as follows.

“Advancements of the model”: The model of Tian Shan formation has evolved to include three main modes: northward underthrusting, double underthrusting, and southward underthrusting models (see Fig. 1 in our main text). These models have been developed and refined through various deep geophysical profiles and geological sections (e.g., Allen et al., 1999 Tectonics; Gao et al., 2013 Lithosphere; Sun et al., 2022 Geology; Zhang et al., 2020 GRL, 2022 Geology; Lei et al., 2007 PEPI; Li et al., 2022 Nat. Commun.). Advances in geophysical imaging have allowed for a more detailed understanding of lithospheric structure. The progression of geophysical imaging technologies continuously enhances our models, reflecting the advancement of research at each stage.

A unique aspect of our work is that we move beyond a simple kinematic model, and interpret the dynamics that drive such a model. Specifically, the integration of aeromagnetic data with other geological and geophysical datasets supports a pure-shear deformational model for the Tian Shan. We take this interpretation further, and use the aeromagnetic information to interpret that the partitioning of iron within the lithosphere is what drives pure-shear mode of deformation.

“Errors in past research”: We respectfully argue that there are no specific errors in past research. Instead, each published research contributes valuable observations that help validate and refine existing models. Our science progresses through improvements on past interpretations with new understandings and updated datasets. As stated above, our work expands on past understandings, but we do offer an entirely new interpretation that suggests that iron partitioning across the lithosphere ultimately controls the kinematics of deformation.

“Validation of the new model”: The reviewer has argued that our model is not

validated by seismic reflection surveys. However, there are no available seismic reflection surveys to properly test our model, and we believe that it is unfair to discard a hypothesis because it cannot be tested by data that does not yet exist. Instead, we systematically have shown how our proposed model supports a wide variety of alternative datasets, including (1) receiver function imaging, (2) thermochronology traverses across the Tian Shan, (3) Quaternary fault studies across the Tian Shan, (4) geodesy velocity fields, and (5) our systematic numerical simulations.

Reviewer #1 also commented: “Otherwise, the so-called model is merely a hypothesis without much evidence, which will not garner widespread interest from readers.” We respectfully disagree with this assessment, as our integrated approach provides substantial evidence for the validity of our model. Our model makes substantial predictions that we test with available datasets, as described in the paragraph above, and will continue to be tested with new field and geophysical studies.

2. The author acknowledges the significant role of gravity models in explaining deep structures, particularly the undulations of the Moho surface. However, there is still a lack of effective gravity data and their processed results to directly support the research in this paper. This ambiguity greatly reduces the reliability of the research findings.

Reply:

We provided a detailed reply to a similar comment in the first round of reviews. It is impossible to differentiate anomalies originating from lithospheric density heterogeneity and Moho topography if there is no seismic-based Moho constraint. There is no high-resolution gravity and seismic velocity data across the whole Tian Shan. Therefore, we have no way to improve the manuscript based on this review comment. Most importantly, the lithosphere-scale receiver function imaging (Li et al., 2022 NC; Zhang et al., 2021 Geology) clearly reveals the Moho morphology of Tian Shan (Fig. 6). The Moho morphology is the key constraint for validating the model.

[REDACTED]

Fig. 6 Lithosphere-scale receiver function imaging
(Zhang et al., 2021 *Geology*, Fig. c; Li et al., 2022, Fig. d)

3. Moreover, from the results of multiple numerical simulations with changed parameters provided by the author, it seems to further confirm the previous doubts, indicating the uncertainty of the model. Each mode obtained from the numerical simulations is different, and the discrepancies are quite significant.

Reply:

We politely argue that this review comment does not provide any specifics. We systematically tested the parameter sensitivity at the request of both reviewers in the initial round of reviews. The additional numerical experiments together with the original ones presented in the text demonstrate that the lithosphere mantle density and associated crustal composition (and thereby rheology) reflected by spatial variations in the Fe content are two critical factors controlling the style of orogenic deformation.

4. It is suggested that the author continue to conduct in-depth research, using observational data, particularly deep reflection seismic data, as the primary constraint conditions for the structure. Geological, geophysical data, including gravity and magnetic data, should be integrated into a rigorous model framework, preferably a quantifiable one. Such research results would be valuable. Speculations based on certain aspects in this paper are not recommended as they do not solve the problem but

rather introduce another possibility, which brings more potentially insignificant issues to the study of the Tianshan orogenic belt.

In all, this version of the manuscript can not be accepted for publication.

Reply:

As we have addressed above, we politely argue against this comment. We have provided a multi-disciplinary dataset to support our model, which we test against numerical models and published geophysical and geological observations. The reviewer does not point out specific disagreements or flaws, but rather provides subjective statements.

We found the rejection review did not consider our detailed response to the first review. Please see our responses to this new review below, and please note that it had no specific objections to our datasets, analyses, or interpretations. Perhaps based on the Reviewer #1's expertise, we believe they downplayed the importance of aeromagnetic imaging and exaggerated the importance of seismic reflection approach, ignoring its limitations.

We also found the latest review comments subjective, especially after our strong efforts to systematically respond to and address comments from the first round of reviews. Reviewer #1 raised the same comments in the second round of review as in the first round, which led us to believe that Reviewer #1's viewpoint was not aligned with the results of our paper due to personal bias, or otherwise Reviewer #1 may have conflict of interest with our research.

Therefore, we believe that Reviewer 1 #1 is not suitable to serve as a reviewer for this manuscript.

Below, please find our response to the first round of reviews (NCOMMS-24-13052-T)

Reviewer #1 (Remarks to the Author):

This study tries to investigate mode of intracontinental mountain building by using Tianshan orogenic belt as an example. However, my opinion seems to be not supportive. Because there are so many critical issues should be addressed.

1. One of the biggest problems in this paper is that there are already a lot of deep reflection seismic data including the book of Xiao Y.C., Liu, X., and Gao, R. et al. (2004) about the Tianshan orogenic belt, which can clearly reveal the orogenic pattern, but the authors seem to ignore it.

Reply:

We appreciate the reviewer's comment regarding existing literature and geophysical surveys across the Tian Shan. Here we acknowledge that there have been decades of important high-resolution geophysical investigations across the Tian Shan (Figs. R1a and R1b), including the reference Xiao, X.C., Liu, X., and Gao, R. et al. (2004) provided by this reviewer. This specific reference is a Chinese book entitled "The crustal structure and tectonic evolution of Southern Xinjiang, China" (Xiao X.C, Liu X., and Gao R. et al., 2004). We attach the copy of this book in the NC submission system for the reviewers and editors to further review and evaluate (the Related Manuscript File).

While working on this project, we have reviewed and compiled all available deep reflection seismic profiles across the Tian Shan, including Xiao, X.C. et al. (2004), to synthesize the competing Tian Shan construction models. The stated models, including the northward underthrusting, double underthrusting, and southward underthrusting models (corresponding to the Fig. 1 in our main text), are based on deep geophysical profiles and some geological sections (e.g., Allen et al., 1999 Tectonics; Gao et al., 2013 Lithosphere; Sun et al., 2022 Geology; Zhang et al., 2020 GRL, 2022 Geology; Lei et al., 2007 PEPI; Li et al., 2022 NC). The references we cited in the manuscript are high-quality geological and geophysical sections, such as the deep seismic section across the Tian Shan (Fig. R2) (Li Wei et al., 2022 NC). Conversely, this stated reference (Xiao, X.C., Liu, X., and Gao, R. et al., 2004) is a book that is in only available in Chinese and not as accessible for international scientists. Therefore, while originally writing this

manuscript, we felt that it was not the most suitable reference for a paper like this submitted to the international, and broad-readership Nature Communications.

Our original writing and presentation of models for the orogenic architecture included a comprehensive review of all possibilities, including the one presented in the Xiao et al. (2004) book. However, we agree with this reviewer here that this is an important reference, as well as other important papers (e.g., Gao et al., 2013 Lithosphere), and therefore we now cite them in our setup of existing work.

We do fully appreciate that such uncited literature and highlighted comments are important to consider. Based on the P-wave velocity structure on lithospheric scale (Fig. R3a), the authors of this book proposed a double-underthrusting model that Junggar and Tarim lithosphere underthrust beneath the Tian Shan orogen by face to face (Fig. R3b). In the revised manuscript, we have now added two additional citations (Xiao et al., 2004 Book; Gao et al., 2013 Lithosphere), especially related to our Fig.1d in the main text (double-size underthrusting beneath the Tian Shan).

Furthermore, during this revision process we have carefully checked the geophysical data and observational figures from Xiao et al. (2004). During this evaluation process, we note only a few deep reflection profiles (i.e., two seismic sections) in the book. All the geophysical text and figures within the book involve in the pages from Page 2 to Page 53, as well as the supplementary figures at the end. We have pasted this information below, and now better cite this study in the main manuscript. Thank you.

[REDACTED]	[REDACTED]
Fig.R1a Location of deep seismic profiles in the West Kunlun-Tarim-Tian Shan domain	Fig.R1b Crustal structure model in the West Kunlun-Tarim-Tian Shan domain

[REDACTED]	
Fig.R2 Deep seismic section across the Tian Shan (Li Wei et al., 2022 NC)	
[REDACTED]	[REDACTED]
Fig.R3a Tomographic image of P waves across the Junggar-Tian Shan-Tarim domain	Fig.R3b Double-underthrusting model proposed by the authors in this book (Xiao, Liu and Gao et al., 2004)

[REDACTED]

[REDACTED]

[REDACTED]

[REDACTED]

[REDACTED]

[REDACTED]

[REDACTED]	[REDACTED]
---	--

[REDACTED]	[REDACTED]
[REDACTED]	[REDACTED]

[REDACTED]	[REDACTED]
[REDACTED]	[REDACTED]

[REDACTED]	[REDACTED]
---	---

[REDACTED]

[REDACTED]

[REDACTED]

[REDACTED]

[REDACTED]

[REDACTED]

[REDACTED]

[REDACTED]

2. The other major issue with the paper is regarding the interpretation of aeromagnetic data, which happens to be a crucial piece of evidence supporting the final conclusions of this study. The authors attempt to use the relationship between aeromagnetic data and Fe-bearing minerals in the crust to explain the tectonic mechanism. However, based on my long-term experience and deep understanding, such a relationship is evidently questionable. This is because aeromagnetic anomalies that could reflect such large-scale, deep-seated (crustal-scale) mineral magnetic anomalies would necessarily be low-intensity long-wavelength components, which are not demonstrated or reflected in the data processing presented in the paper. Regional aeromagnetic anomalies typically reflect regional metamorphic basement (including metamorphic magnetite, etc.), which is a basic global pattern. From a geophysical perspective, it is recommended to use gravity data for applications that aim to reflect large-scale orogenic styles because the long-wavelength anomalies in Bouguer gravity data reflect changes at the crust-mantle boundary, which may explain orogenic events more effectively than aeromagnetic data.

Reply:

We appreciate the reviewer's important comments about the interpretation of aeromagnetic anomaly impacting the modes of mountain building. We agree with reviewer #1's viewpoints that "*.....aeromagnetic anomalies that could reflect such large-scale, deep-seated (crustal-scale) mineral magnetic anomalies would necessarily be low-intensity long-wavelength components.....*". From a geophysical perspective, there are at least two methods that can be applied to delineate magnetic sources at various depths and scales. One method involves potential field conversion, which encompasses derivation and upward continuation. Derivation can emphasize shallow magnetic source (Blackly, 1996), whereas the upward continuation can effectively delineate deeper magnetic source, such as the metamorphic basement and mid-lower crust (e.g., Blackly, 1996; Xu et al., 2023 Geology). The other method is magnetic inversion, which can directly provide the magnetization intensity (magnetic susceptibility) at any depths above the Curie surface (e.g., Hu et al., 2019 Geophysics; Xu et al., 2023 Geology). Magnetic inversion is a well-established and well-proven method that has been widely utilized to delineate the magnetic structure of the whole crust (e.g., Xu et al., 2023 Geology), even the uppermost lithospheric mantle (Ferre et al., 2021 Nature REE).

In our manuscript, we used upward continuation on the magnetic grid data (Fig. S5 in our Supplementary Material file; also pasted below). The magnetic anomalies processed by upward continuation at various altitude still exhibit low values in the Tian

Shan, suggestive of weak magnetic properties of the deep crust in this orogenic belt. Furthermore, the 3D magnetic inversion was also utilized (Fig. S6 in our Supplementary Material file; also pasted below), which revealed that the middle-lower crust of the Tian Shan is characterized by weak magnetization intensity (Fig. S6). Therefore, through these two data-processing methods, we can effectively decipher the information about large-scale, deep-seated “mineral magnetic anomalies” and calculated the Fe content via the empirical correlation between the magnetic susceptibility and the magnetite (Fe_3O_4) percentage (Grant, 1985 Geoexploration).

Fig. S5 Map of magnetic anomalies at different upward continuation values to enhance regional and deep magnetic features. Panels a through f correspond to upward continuation of 1 km, 5 km, 10 km, 20 km, 30 km and 40 km, respectively.

Fig. S6

Fig. S6 Crust-scale magnetization intensity model for the Tian Shan and its bounding cratons.

Here we wish to emphasize that we have also provided details of our processing methods in the Supplementary Material file.

The reviewer #1 noted that “...Regional aeromagnetic anomalies typically reflect regional metamorphic basement (including metamorphic magnetite, etc.), which is a basic global pattern...”. According to classic geomagnetic literature (e.g., Krutikhovskaya and Pashkevich, 1979 *Journal of Geophysics*; Frost and Shive, 1986 *JGR*; Shive, 1989 *GRL*), regional (high-amplitude, long-wavelength) aeromagnetic anomalies reflect contributions from local metamorphic basement but also the middle-lower crust as well as the upper mantle (Friedman et al., 2014 *Tectonophysics*; Ferre et al., 2021 *Nature REE*). Moreover, for example, the 3-D inversion of the aeromagnetic data can reveal the vertical and lateral compositions at the crustal scale (0-40 km).

(Kolawole et al., 2017 GRL), providing key information to decipher the mechanism of great earthquake generation. Therefore, aeromagnetic anomalies have proven to be effective data in imaging the whole crustal structure.

We also concur with reviewer #1's comment: “.....*long-wavelength anomalies in Bouguer gravity data reflect changes at the crust-mantle boundary.....*”. Bouguer gravity data can be used to assess Moho topography, particularly in regions lacking seismic reflection and receiver function survey data. However, there are at least two challenges when interpreting the orogenic structures from such data. First, both lithospheric density heterogeneity and Moho topography can contribute to the observed gravity anomaly (Mooney and Kaban, 2010 JGR; Wang et al., 2014 Tectonophysics). It is impossible to differentiate anomalies originating from lithospheric density heterogeneity and Moho topography if there is no seismic-based Moho constraint (Guo et al., 2019 EPSL). Therefore, determining the Moho from only Bouguer gravity can be very uncertain. Second, even if Moho topography is relatively accurately determined from Bouguer gravity, it is still challenging to determine the mechanism behind Moho topography, as discrete continental subduction, distributed pure-shear shortening, and

ductile flow in the middle and/or lower crust can all influence Moho topography. Therefore, here we did not utilize gravity data to investigate the crust-mantle boundary due to its uncertainty and limited relevance to our research theme. We welcome other studies that attempt such methods, but we did not feel that this would be the best approach to address our research goals. In contrast, magnetic anomalies are sensitive to Fe-bearing minerals, and the presence of Fe-bearing minerals affects lithospheric density distribution and viscous rheology, which are key factors influencing the mechanism of intracontinental mountain building.

We appreciate this discussion to sharpened our text.

Thank you.

- Blakely, R. J. (1996). Potential theory in gravity and magnetic applications. Cambridge university press.
- Frost, B. R., & Shive, P. N. (1986). Magnetic mineralogy of the lower continental crust. *Journal of Geophysical Research: Solid Earth*, 91(B6), 6513-6521.
- Friedman, S. A., Feinberg, J. M., Ferré, E. C., Demory, F., Martín-Hernández, F., Conder, J. A., & Rochette, P. (2014). Craton vs. rift uppermost mantle contributions to magnetic anomalies in the United States interior. *Tectonophysics*, 624, 15-23.
- Ferré, E. C., Kuppenko, I., Martín-Hernández, F., Ravat, D., & Sanchez-Valle, C. (2021). Magnetic sources in the Earth's mantle. *Nature Reviews Earth & Environment*, 2(1), 59-69.
- Grant, F. S. (1985). Aeromagnetism, geology and ore environments, I. Magnetite in igneous, sedimentary and metamorphic rocks: an overview. *Geoscientific Exploration*, 23(3), 303-333.
- Hu, M., Yu, P., Rao, C., Zhao, C., & Zhang, L. (2019). 3D sharp-boundary inversion of potential-field data with an adjustable exponential stabilizing functional. *Geophysics*, 84(4), J1-J15.
- Kolawole, F., Atekwana, E. A., Malloy, S., Stamps, D. S., Grandin, R., Abdelsalam, M. G., ... & Shemang, E. M. (2017). Aeromagnetic, gravity, and Differential Interferometric Synthetic Aperture Radar analyses reveal the causative fault of the 3 April 2017 Mw 6.5 Moiyabana, Botswana, earthquake. *Geophysical Research Letters*, 44(17), 8837-8846.
- Krutikhovskaya, Z. A., & Pashkevich, I. K. (1979). Long-wavelength magnetic anomalies as a source of information about deep crustal structure. *Journal of Geophysics*, 46(1), 301-317.
- Mooney, W. D., & Kaban, M. K. (2010). The North American upper mantle: Density, composition, and evolution. *Journal of Geophysical Research: Solid Earth*, 115(B12).
- Shive, P. N. (1989). Can remanent magnetization in the deep crust contribute to long wavelength magnetic anomalies?. *Geophysical Research Letters*, 16(1), 89-92.
- Xu, X., Chen, H., Zusa, A. V., Yin, A., Yu, P., Lin, X., ... & Wang, B. (2023). Phanerozoic cratonization by plume welding. *Geology*, 51(2), 209-214.

3. The authors tried to validate the orogenic model using numerical simulation methods, but a major flaw is that the results of numerical simulations are highly dependent on initial and boundary conditions. However, this study fails to analyze the parameter sensitivity of these simulation results. This lack of analysis means that the conclusions drawn may not necessarily be universally applicable or stable, but rather specific to certain conditions or scenarios.

Reply:

We appreciate and acknowledge the comments the reviewer brought up here. For these types of numerical simulations, the initial and boundary conditions are critical constraints for how the models will respond to parameter variations. In the past decade, series of numerical and physical modeling tests suggest that the lithospheric-scale rheology and density (lower crust and mantle lithosphere) primarily control the lithosphere deformation and structure in the intracontinental tectonism (e.g., Calignano et al., 2015 *Tectonics*; Chen et al., 2017 *Nature Communications*; Huangfu et al., 2021 *Nature Communications*).

In this work, we first provide multi-method geological-geophysical data that we interpret shows that lower crustal composition and mantle lithosphere depletion (rheology and density) control the mode of intracontinental mountain building. Our primary data revolves around these geological and geophysical datasets. From these datasets, we construct a hypothesis that the partitioning of lithospheric iron can simultaneously influence lithospheric density, rheology, and magnetic structure to influence the orogenic mode. We tested this hypothesis with a series of numerical simulations. Therefore, here we wish to point out that our interpretations are not uniquely dependent on the numerical simulations, but rather the geophysical and geologic observations. The numerical simulations serve to test the validity and viability of our hypothesis.

That said, we appreciate the fact that any numerical simulations are highly dependent on the initial and boundary conditions. The parameter sensitivity analysis is indeed very important pre-condition for numerical modeling and geological interpretation. We have therefore conducted additional parameter sensitivity tests including 20 additional models, which are now presented in the supplementary material (Table S3; Such table screenshot pasted below) and discussed in the revised text. In these models, we keep the initial thermal structure of the bounding cratons the same as

that of the orogen, and test the influences of the depletion density of the craton mantle and the lower crustal rheology with different compositions. The new results show that the depletion-induced density reduction of the cratonic mantle lithosphere is a key factor controlling the style of crustal deformation at the collisional zones (Fig. R4). It can also be seen that the models with mafic lower crust exhibit obvious simple shear behavior (Fig. R5a), whereas the ones with felsic lower crust have limited lower crustal underthrusting, behaving as pure shear (Fig. R5b). The additional numerical experiments together with the original ones presented in the text demonstrate that the lithosphere mantle density and associated crustal composition (and thereby rheology) reflected by spatial variations in the Fe content are two critical factors controlling the style of orogenic deformation.

Table S3

Table S3 Parameters and Results of Conducted Experiments

Model name	Orogen Moho temperature (°C)	Craton Moho temperature (°C)	Orogen lower crust flow law	Craton lower crust flow law	Depletion density ^a (kg/m ³)	Deformation Mode
TSM01	700	600	Felsic granulite	Mafic granulite	30	Pure shear
TSM02	600	600	Felsic granulite	Mafic granulite	30	Simple shear
TSM03	600	600	Felsic granulite	Felsic granulite	30	Pure shear
TSM04	600	600	Mafic granulite	Mafic granulite	30	Simple shear
TSM05	700	600	Felsic granulite	Mafic granulite	0	Simple shear
TSM06	600	600	Felsic granulite	Mafic granulite	0	Simple shear
TSM07	600	600	Felsic granulite	Felsic granulite	0	Pure Shear
TSM08	600	600	Mafic granulite	Mafic granulite	0	Simple shear
TSM09	600	600	Felsic granulite	Mafic granulite	60	Pure Shear
TSM10	600	600	Felsic granulite	Felsic granulite	60	Pure Shear
TSM11	600	600	Mafic granulite	Mafic granulite	60	Simple shear
TSM12	600	600	Felsic granulite	Mafic granulite	-30	Simple shear
TSM13	600	600	Felsic granulite	Felsic granulite	-30	Pure Shear
TSM14	600	600	Mafic granulite	Mafic granulite	-30	Simple shear
TSM15	600	600	Felsic granulite	Mafic granulite	-60	Simple shear
TSM16	600	600	Felsic granulite	Felsic granulite	-60	Pure Shear
TSM17	600	600	Mafic granulite	Mafic granulite	-60	Simple shear

^a A negative depletion density means that the lithospheric mantle of the craton is denser than that of the orogen.

Table S3 in the supplementary material file

Fig. R4 Geodynamic modeling of the role of lithospheric depletion on the orogeny style. Collision between two continents with iron-fertile and iron-depleted mantle density varying from $\Delta\rho=+30$ to -60 kg/m^3 . The models with lithospheric mantle depletion-induced density contrast of -60 kg/m^3 (a) and -30 kg/m^3 (b) lead to pure shear-style thickening of an orogen via tectonic wedging, whereas 0 kg/m^3 (c) and $+30$ kg/m^3 (d) leads to underthrusting / subduction of continental lithosphere under the orogen.

Here, it is expressed as $\Delta\rho_m = \rho_{OLM} - \rho_{CLM}$, where ρ_{OLM} and ρ_{CLM} represent the reference density of the orogen and continent lithospheric mantle, respectively.

The lower crust of the continental and orogenic lithosphere are equipped with mafic and felsic rocks, respectively.

Fig. R5 Geodynamic modeling of the role of lower crustal composition on the orogeny style. Collision between two continents with mafic (a) and felsic (b) orogenic lower crust. Orogen-bounding continents are here equipped with mafic lower crust. The models are equipped with lithospheric mantle depletion density of -30 kg/m^3 .

The addition of these sensitivity tests will be valuable to future readers to assess controls on the styles of orogeny. Thank you for this valuable suggestion. We do note that due to length limitations for the journal, we chose to only display the represent model results in the main text of the manuscript.

Thank you.

Responses to Reviewers

MS Title: Mode of intracontinental mountain building controlled by lower crustal composition and mantle lithosphere depletion

Tracking #: NCOMMS-24-13052B-Z

This document is our reply and discussion of these review comments. Please note that below we provide detailed responses in *blue font* to each of these comments.

#####

Reviewer #3 (Remarks to the Author):

Peer Review of

Mode of intracontinental mountain building controlled by lower crustal composition and mantle lithosphere depletion

Manuscript #NCOMMS-24-13052A

submitted to Nature Communications

This study presents the results of an aeromagnetic study of the Tian Shan orogen and the bounding Tarim Craton and Kazakh Craton. The data reveals different lithospheric composition between the Tian Shan belt and the surrounding cratons: The former has a more felsic lower crust and a less depleted mantle lithosphere, while the latter possesses a more depleted lithospheric keel and a more mafic lower crust. The authors argue that, as a result of the compositional difference, the strong and buoyant cratons resist internal deformation and subduction. The weak Tian Shan belt, on the other hand, deforms by distributed pure shearing. To test this hypothesis, geodynamic modeling is carried out with varying depletion-induced mantle lithospheric density, lower crustal compositions and thermal conditions. The models show that pure shear deformation mode is favored if the less depleted lithospheric mantle and more felsic lower crust are present in the orogenic belt. Otherwise, the crustal deformation is dominated by simple shearing mode or subduction of the bounding craton. Furthermore, the high temperature thermal state also favors distributed pure shearing deformation mode, although the thermal state per se may only play a minor role in controlling the crustal deformation in the Tian Shan belt.

I find the multidisciplinary approaches the authors have taken to formulate and test their tectonic hypothesis interesting and the implications of the study concerning the crustal deformation are of global significance. I notice that geophysical data have been largely used to propose the tectonic model, though there are still some imperfect results that

can be done in the future study. To me, the manuscript is generally well-organized and written and is worthwhile for publication in the journal. I do suggest a few improvements on the chain of logic that leads to the final conclusions. I thereby support publication of the manuscript with minor revisions. Some major comments are as follows:

Thank you for the positive feedback on our manuscript. We carefully read through these comments and have modified the manuscript and figures accordingly. These latest changes have further refined our paper. In particular, the suggestions regarding hot Precambrian orogeny allow us to add some more breadth to the final discussion paragraph of our paper. Consideration of this point results in a self-consistent explanation for Precambrian orogeny and cratonic growth. Thank you for these suggestions.

1. I don't recommend using the term 'Kazakh Craton'. We know that the 'craton' has a clearly geological definition. I would suggest using the term 'Kazakhstan Block' in the text.

We greatly appreciate and acknowledge the constructional suggestion.

We have accordingly used the term 'Kazakhstan Block' in the text, and further revised the term 'Tarim Craton' to be 'Tarim Block'. Thank you for the suggestion. We agree that using block is a more generalized term for these pieces of continent that have remained rigid during the Mesozoic-Cenozoic time.

2. The numerical modeling results presented in Table S3 have an alternative interpretation. In model names TSM03, TSM07, TSM10, TSM13 and TSM16, the lower crustal flow law for both the orogenic the cratonic lower crust are felsic granulite and they all exhibit pure shear deformation mode. In contrast, in model names TSM04, TSM08, TSM11, TSM14 and TSM17, the lower crustal flow laws are both mafic granulite and they all display simple shear mode. This leads to the alternative interpretation that the deformation mode is strongly controlled by the lower crust rheology rather than the density (or viscosity) contrast caused by the differential depletion between the orogenic and cratonic lithosphere. This contradicts with the statement made in Lines 235-237, in the sense that the orogeny can still undergo pure shearing even when there is "no differences in lower crustal rheology and lithospheric mantle density between the orogen and its bounding continents". We suggest the authors to provide further clarification on this in the supplementary materials if not in the main text.

We greatly appreciate this concern so that we can better clarify the text of these

modeling results and present our main points.

For numerical simulation, the initial and boundary conditions are critical constraints for how the models will respond to parameter variations. In the past decade, a series of numerical and physical modeling tests suggest that the lithospheric-scale rheology and density (lower crust and/or mantle lithosphere) primarily control the lithosphere deformation and structure in the intracontinental tectonism (e.g., Calignano et al., 2015 *Tectonics*; Chen et al., 2017 *Nature Communications*; Huangfu et al., 2021 *Nature Communications*). By comparing these pure-shear (TSM03, TSM07, TSM10, TSM13 and TSM16) and (TSM04, TSM08, TSM11, TSM14 and TSM17) models, it is highlighted by the reviewer that the deformation mode is mainly dominated by the lower crust rheology.

In this work, we first provided multi-method geological-geophysical data and interpreted the model that lower crustal composition and mantle lithosphere depletion (rheology and density) control the mode of intracontinental mountain building. Our primary data revolves around these geological and geophysical datasets. From these datasets, we constructed a hypothesis that the partitioning of lithospheric iron can simultaneously influence lithospheric density, rheology and magnetic structure to influence the orogenic mode. We tested this hypothesis with a series of numerical simulations. Therefore, here we would like to point out that our interpretations are not uniquely dependent on the numerical simulations, but rather the geophysical and geological observations. The numerical simulations serve to test the validity and viability of our hypothesis.

The parameter sensitivity analysis is very important pre-condition for numerical modeling and geological interpretation. We therefore conducted additional parameter sensitivity tests including 20 additional models, which are now presented in the supplementary material (Table S3; Such table screenshot pasted below) and discussed in the original text. In these models, we kept the initial thermal structure of the bounding cratons the same as that of the orogen, and tested the influences of the depletion density of the craton mantle and the lower crustal rheology with different compositions. The new results show that the depletion-induced density reduction of the cratonic mantle lithosphere is a key factor controlling the style of crustal deformation at the collisional zones (Fig. S1). It can also be seen that the models with mafic lower crust exhibit obvious simple shear behavior (Fig. S2a), whereas the ones with felsic lower crust have limited lower crustal underthrusting, behaving as pure shear (Fig. S2b). The additional numerical experiments together with the original ones presented in the text demonstrate that the lithosphere mantle density and associated crustal composition (and thereby rheology) reflected by spatial variations in the Fe content are the two critical factors

controlling the style of orogenic deformation.

The addition of these sensitivity tests will be valuable to readers to assess controls on the styles of deformation and orogeny in various tectonic settings. Thank you for this valuable suggestion. We do note that due to length limitations for the journal, we chose to only display the representative model results in the main text of the manuscript.

Finally, we have further modified the text to provide the clarification in the supplementary materials, as the reviewer highlighted. Thank you.

Table S3 Parameters and results of conducted numerical experiments

Model name	Orogen Moho temperature (°C)	Craton Moho temperature (°C)	Orogen lower crust flow law	Craton lower crust flow law	Depletion density ^a (kg/m ³)	Deformation Mode
TSM01	700	600	Felsic granulite	Mafic granulite	30	Pure shear
TSM02	600	600	Felsic granulite	Mafic granulite	30	Simple shear
TSM03	600	600	Felsic granulite	Felsic granulite	30	Pure shear
TSM04	600	600	Mafic granulite	Mafic granulite	30	Simple shear
TSM05	700	600	Felsic granulite	Mafic granulite	0	Simple shear
TSM06	600	600	Felsic granulite	Mafic granulite	0	Simple shear
TSM07	600	600	Felsic granulite	Felsic granulite	0	Pure Shear
TSM08	600	600	Mafic granulite	Mafic granulite	0	Simple shear
TSM09	600	600	Felsic granulite	Mafic granulite	60	Pure Shear
TSM10	600	600	Felsic granulite	Felsic granulite	60	Pure Shear
TSM11	600	600	Mafic granulite	Mafic granulite	60	Simple shear
TSM12	600	600	Felsic granulite	Mafic granulite	-30	Simple shear
TSM13	600	600	Felsic granulite	Felsic granulite	-30	Pure Shear
TSM14	600	600	Mafic granulite	Mafic granulite	-30	Simple shear
TSM15	600	600	Felsic granulite	Mafic granulite	-60	Simple shear
TSM16	600	600	Felsic granulite	Felsic granulite	-60	Pure Shear
TSM17	600	600	Mafic granulite	Mafic granulite	-60	Simple shear

^a A negative depletion density means that the lithospheric mantle of the craton is denser than that of the orogen.

Table S3 The parameter sensitivity analysis

orogeny style. Collision between two continents with iron-fertile and iron-depleted mantle density varying from $\Delta\rho=+30$ to -60 kg/m³. The models with lithospheric mantle depletion-induced density contrast of -60 kg/m³ (a) and -30 kg/m³ (b) lead to pure shear-style thickening of an orogen via tectonic wedging, whereas 0 kg/m³ (c) and $+30$ kg/m³ (d) leads to underthrusting / subduction of continental lithosphere under the orogen.

Here, it is expressed as $\Delta\rho_m = \rho_{OLM} - \rho_{CLM}$, where ρ_{OLM} and ρ_{CLM} represent the reference density of the orogen and continent lithospheric mantle, respectively.

The lower crust of the continental and orogenic lithosphere are equipped with mafic and felsic rocks, respectively.

Fig. S2 Geodynamic modeling of the role of lower crustal composition on the orogeny style. Collision between two continents with mafic (a) and felsic (b) orogenic lower crust. Orogen-bounding continents are here equipped with mafic lower crust. The models are equipped with lithospheric mantle depletion density of -30 kg/m³.

3. The authors used the $\Delta\rho$ value as a proxy for depletion-induced density variation and tested models with $\Delta\rho$ in -60, -30, 0 and 30 kg/m³. We suggest the authors to provide justification as to why these values are chosen and whether they are geologically reasonable.

Thanks for the above comment so that we can provide justification and better clarify the text.

The process of partial melting of mantle peridotite leads to systematic changes in the nature of the peridotite with extent of melt removal, transforming an un-melted or “fertile” peridotite into a residual or “depleted” peridotite, accompanied by changes in lithology, mineralogy, mineral chemistry and density (Pearson et al., 2021 Nature).

Here, it is expressed as $\Delta\rho = \rho_{olm} - \rho_{clm}$, where ρ_{olm} and ρ_{clm} represent the reference density of the orogen and continent lithospheric mantle, respectively. As shown in the figure presented above, these $\Delta\rho$ values in -60, -30, 0, +30 and +60 kg/m³ are corresponding to the percentage change in density spanning -2.0% to +2.0%). The reference density of lithospheric mantle is 3300 kg/m³. Thus, these $\Delta\rho$ values of -60, -30, and 0 kg/m³ are approximately equal to $-2.0\% \times 3300 \text{ kg/m}^3 (\approx -60)$, $-1.0\% \times 3300 \text{ kg/m}^3 (\approx -30)$, $-0.0\% \times 3300 \text{ kg/m}^3 (\approx 0)$ and $+1.0\% \times 3300 \text{ kg/m}^3 (\approx +30)$.

The reference density for the orogenic lithospheric mantle is fixed at 3300 kg/m³. The more positive $\Delta\rho$ values correspond to greater percentage melting of lithospheric mantle peridotite that a continent lithosphere experienced, whereas a negative $\Delta\rho$ value

geologically corresponds to metasomatism acting on the continent lithospheric mantle peridotite (Fig.S1). In this sense, each $\Delta\rho$ value is marked by a corresponding percentage melting/metasomatism value.

We have further modified the relevant text if it was not that clear before.

4. The authors discuss the impact of thermal state on the crustal deformation behavior from a geodynamic modeling perspective (Figure. S16). Although the authors preclude the thermal state as a major contributing factor to the crustal deformation in the Tian Shan belt (Lines 286- 291), We suggest the authors to briefly touch upon the role of thermal state on Precambrian orogeny (e.g. Chardon et al., 2009) and by doing this, generalize the applicability of the current modeling work to a wider range of geological time.

References

D. Chardon, D. Gapais, and F. Cagnard. Flow of ultra-hot orogens: A view from the Precambrian, clues for the Phanerozoic. *Tectonophysics*, 477(3-4):105–118, Nov. 2009. ISSN 00401951. doi: 10.1016/j.tecto.2009.03.008.

We greatly appreciate and acknowledge the constructional suggestions. These suggestions have led to a few brief, yet important modifications to our manuscript.

First, we use the observations that the hot Precambrian accretionary orogens deform via pure shear to further validate our ancillary numerical simulations in the thermal state section. We had written that these numerical simulations suggest that pure-shear deformation results from hot orogeny, and we can now refer to interpretations of Precambrian accretionary orogens to further support this inference, by citing Chardon et al. (2009).

Second, we expanded the final paragraph to review how our interpretations are consistent with the established interpretations for Precambrian craton growth and continental evolution. Specifically, a hotter Precambrian mantle would result in higher degrees of melting, which would predictably result in generally more melt depletion of the continental mantle lithosphere (e.g., Lee et al., 2011; Xu et al., 2023). A more melt depleted mantle lithosphere coupled with hotter orogeny would predict pure-shear styles of orogeny, as reviewed by Chardon et al. (2009) for Precambrian orogens. Pure-shear orogeny in Archean-Proterozoic would lead to lateral accretionary growth of the continents to build cratons (Pearson et al., 2021), while simultaneously preserving these growing cratons by sparing them from simple-shear continental subduction. Our newly revised paragraph links the interpretations of our orogenic model to continental

evolution in a self-consistent manner, which further strengthens and supports our arguments.

Based on these reviews, we have further modified the main text to cover these points discussed above, and generalized the applicability of our numerical modelling. Thank you for all these suggestions. We hope you are satisfied with our responses and presentation.

Responses to Reviewers

MS Title: Mode of intracontinental mountain building controlled by lower crustal composition and mantle lithosphere depletion

Tracking #: NCOMMS-24-13052D

This document is our reply and discussion of these review comments. Please note that below we provide detailed responses in *blue font* to each of these comments.

#####

Reviewer #3 (Remarks to the Author):

Peer Review of

Manuscript # NCOMMS-24-13052C

Mode of intracontinental mountain building controlled by lower crustal composition and mantle lithosphere depletion

Comments:

The authors have done a thorough improvement addressing all the review comments from the previous round. The manuscript presents a tightly integrated multidisciplinary study combining aeromagnetic analysis and numerical modeling, with profound implications for crustal deformation modes in both Precambrian and Phanerozoic orogenies. In its present form, the manuscript has been significantly improved in terms of logical clarity, as well as the breadth and depth of discussion. Only a few minor, mostly syntactic, edits are needed, so I am happy to recommend acceptance for publication in Nature Communications (NC) after these corrections.

Thank you for the positive feedback on our manuscript. We carefully read through these comments and have modified the manuscript and figures accordingly. These latest changes have further refined our paper. Thank you for these suggestions.

Lines 132-134 Since most of the discussion (and in fact the whole article) is centered around differentiating the pure shear and simple shear modes of deformation for the Tian Shan orogenic belt and the surrounding continental blocks, it would be helpful to introduce these key concepts in the first few paragraphs of the Introduction section.

Thanks for the above comment so that we can provide justification and better clarify the text.

Line 85 repetition of the word “depleted” . . . presence of a depleted iron-poor depleted mantle lithosphere that is rheologically . . .

Thanks. Corrected.

Lines 87-88 . . . mantle lithosphere . . . bounds the two sides of the Tian Shan belt, which is the latter characterized by more felsic iron-poor lower crust and less depleted, thinner mantle lithosphere.

Thanks. Corrected.

Line 98 . . . accelerated deformation in the Miocene and is still being actively shortening shortened today.

Thanks. Corrected.